# NEAR-OPTIMAL ALGORITHMS FOR AUTONOMOUS EXPLORATION AND MUITI-GOAL STOCHASTIC SHORTEST PATH

## ABSTRACT

We revisit the incremental autonomous exploration problem proposed by Lim and Auer (2012). In this setting, the agent aims to learn a set of near-optimal goal-conditioned policies to reach the $L$-controllable states: states that are incrementally reachable from an initial state $s_0$ within $L$ steps in expectation. We introduce three new algorithms with stronger sample complexity bounds than existing ones. Furthermore, we also prove the first lower bound for the autonomous exploration problem. In particular, the lower bound implies that one of our proposed algorithms, Value-Aware Autonomous Exploration, is nearly minimax-optimal when the number of $L$-controllable states grows polynomially with respect to $L$. Key in our algorithm design is a connection between autonomous exploration and multi-goal stochastic shortest path, a new problem that naturally generalizes the classical stochastic shortest path problem. This new problem and its connection to autonomous exploration can be of independent interest.

## 1 INTRODUCTION

Reinforcement learning (RL) with a known state space has been studied in a wide range of settings (e.g., Schmidhuber, 1991; Oudeyer et al., 2007; Oudeyer and Kaplan, 2009; Baranes and Oudeyer, 2009). When the state space is large, it is difficult for a learning agent to discover the whole environment. Instead, the agent can only explore a small portion of the environment. At a high level, we hope that the agent can discover states near the initial state, expand the range of known states by exploration, and learn near-optimal goal-conditioned policies for the known states. Because the agent discovers its known states of the environment incrementally, this learning problem was named Autonomous Exploration (AX) (Lim and Auer, 2012; Tarbouriech et al., 2020).

The autonomous exploration problem generalizes the Stochastic Shortest Path (SSP) problem (Bertsekas et al., 2000) where the agent aims to reach a predefined goal state while minimizing its total expected cost. However, in the autonomous exploration setting, the agent aims to discover a set of reachable states in a large environment and find the optimal policies to reach them. The autonomous exploration formulation is applicable to an increasing number of real-world RL problems, ranging from navigation in mazes (Devo et al., 2020) to game playing (Mnih et al., 2013). For example, in the maze navigation problem, a robot aims to follow a predefined path in an unknown environment, and the robot has to discover and expand the size of regions known to itself autonomously without prior knowledge of the environment. This procedure also resembles some biological learning processes. See Lim and Auer (2012) for more discussions.

**Related Work.** The setting of autonomous exploration was introduced by Lim and Auer (2012), who gave the first algorithm, UcbExplore, with a sample complexity $\widetilde{O}(L^3 S_{(1+\varepsilon)L} A/\varepsilon^3)$. Here $L$ denotes the distance within which we hope the learning agent to discover, $S_{(1+\varepsilon)L}$ denotes the total number of states within distance $(1 + \varepsilon)L$ from the starting state, $A$ denotes the size of the action space, and $\varepsilon$ denotes the error that we can tolerate. Recent work by Tarbouriech et al. (2020) designed the DisCo algorithm with a sample complexity bound $\widetilde{O}(L^3 S_{(1+\varepsilon)L}^2 A/\varepsilon^2)^1$, which improves the $1/\varepsilon$

---

[1] We translate their absolute error $\varepsilon$ to the relative error $\varepsilon L$. Their bound has a refined form $\widetilde{O}(L^3 S_{(1+\varepsilon)L} \Gamma_{(1+\varepsilon)} LA/\varepsilon^2)$ where $\Gamma_{(1+\varepsilon)L}$ is the branching factor which in the worst case is $S_{(1+\varepsilon)L}$.

| Algorithm | Sample Complexity | $\mathbf{AX}_L$ on $\mathcal{S}_L^{\rightarrow}$ | $\mathbf{AX}^*$ on $\mathcal{S}_L^{\rightarrow}$ | $\mathbf{AX}_L$ on $\mathcal{K}$ | $\mathbf{AX}^*$ on $\mathcal{K}$ |
|---|---|---|---|---|---|
| UcbExplore (Lim and Auer, 2012) | $\widetilde{O}(L^3 S_{(1+\varepsilon)L} A/\varepsilon^3)$ | Yes | No | Yes | No |
| DisCo (Tarbouriech et al., 2020) | $\widetilde{O}\left(L^3 S_{(1+\varepsilon)L}^2 A/\varepsilon^2\right)$ | Yes | Yes | Yes | Yes |
| VOISD | $\widetilde{O}(L^3 S_{(1+\varepsilon)L} A/\varepsilon^2)$ | Yes | No | Yes | No |
| VOISD + Re-MG-SSP | $\widetilde{O}\left(L^3 S_{(1+\varepsilon)L} A/\varepsilon^2\right)$ | Yes | Yes | Yes | Yes |
| VALAE | $\widetilde{O}\left(L S_{2L} A/\varepsilon^2\right)$ | Yes | Yes | No | Yes |
| Lower Bound | $\Omega(L S_L A/\varepsilon^2)$ | Yes | Yes | Yes | Yes |

Table 1: Comparisons between our results and prior results. Algorithms and results in this paper are in grey cells. We define $L$, $A$, $S_L$, $S_{(1+\varepsilon)L}$, $S_{2L}$ in Sect. 2, and $\varepsilon$ is the target accuracy. We study four criteria, $\mathbf{AX}_L$ **on** $\mathcal{S}_L^{\rightarrow}$, $\mathbf{AX}^*$ **on** $\mathcal{S}_L^{\rightarrow}$, $\mathbf{AX}_L$ **on** $\mathcal{K}$, and $\mathbf{AX}^*$ **on** $\mathcal{K}$ whose definitions are in Sect. 2. For simplicity, for each result, we only display the leading term in terms of the scaling in $1/\varepsilon$.

dependency at the cost of a worse dependency on $S_{(1+\varepsilon)L}$. In this paper, we present new algorithms to further improve the sample complexity.

## 1.1 CONTRIBUTIONS

In this paper, we take important steps toward resolving the autonomous exploration problem. We compare our results with prior ones in Table 1.[2] and we summarize our contributions below:

1. We propose a new state discovery algorithm, Value-Optimistic Incremental State Discovery (VOISD), which uses a value-optimistic method (Neu and Pike-Burke, 2020) to estimate the expected cost of the optimal policy to reach these states. We prove this algorithm enjoys an $\widetilde{O}\left(L^3 S_{(1+\varepsilon)L} A/\varepsilon^2\right)$ sample complexity, which improves prior results in Lim and Auer (2012) and Tarbouriech et al. (2020).
2. We connect the autonomous exploration problem to a new problem, multi-goal SSP and propose a new algorithm Re-MG-SSP that 1) satisfies a stronger criterion[3] and 2) serves as a burn-in step for the next algorithm.
3. We further propose Value-Aware Autonomous Exploration (VALAE), which uses VOISD and Re-MG-SSP as initial steps and then uses the estimated value functions to guide our exploration. By doing so, for each state-action pair $(s, a)$, we derive an $(s, a)$-dependent sample complexity bound, which can exploit the variance information, and yield a sharper sample complexity bound than the bounds for VOISD and Re-MG-SSP. In particular, VALAE improves the dependency on $L$ from cubic to linear.
4. We give the first lower bound of the autonomous exploration problem. This lower bound shows VALAE is nearly minimax-optimal when the $S_L$ grows polynomially with respect to $L$.

## 1.2 MAIN DIFFICULTIES AND TECHNIQUE OVERVIEW

While our work borrows ideas from prior work on autonomous exploration (Lim and Auer, 2012; Tarbouriech et al., 2020) and recent advances in SSP (Tarbouriech et al., 2021), we develop new techniques to overcome additional difficulties that are unique in autonomous exploration.

**Dependence between the Estimated Transition Matrix and Discovered States.** Our algorithm incrementally adds new states to the set of discovered states $\mathcal{K}$. To obtain a tight dependency on $S_{(1+\varepsilon)L}$, similar to the standard RL setting, one needs to use concentration on $(\widehat{P}_{s,a} - P_{s,a})V_{\mathcal{K},g}^*$ instead of $\|\widehat{P}_{s,a} - P_{s,a}\|_1$ (used by Tarbouriech et al. (2020)) where $\widehat{P}_{s,a}$ is the estimated transition,

---

[2]In (Lim and Auer, 2012) and (Tarbouriech et al., 2020), the cost is 1 uniformly for all state-action pairs. In this paper, we allow non-uniform costs. In Table, we consider uniform cost in Table 1 for fair comparisons.

[3]See Sect. 2 for different criteria.

$P_{s,a}$ is the true transition, and $V^*_{\mathcal{K},g}$ is the value function of the optimal policy going to the state $g$ restricted on the discovered states $\mathcal{K}$. The main challenge is that the set of discovered states $\mathcal{K}$ is dependent on the samples collected, and thus $V^*_{\mathcal{K},g}$ is dependent on $\widehat{P}_{s,a}$. We can use the union bound on all the possible $\mathcal{K}$, but the number of possible $\mathcal{K}$ is exponential in the number of states $S$.

Our main technique is to construct a series of sets of states $\{s_0\} = \mathcal{K}_0 \subseteq \mathcal{K}_1 \subseteq \cdots \subseteq \mathcal{K}_Z = \mathcal{S}^{\rightarrow}_L$, where $\mathcal{K}_{z+1}$ is constructed by adding all the states that are reachable from $s_0$ by some policy on $\mathcal{K}_z$ with expected cost no more than $L$. This series is only polynomially large so after applying the union bound we only have a logarithmic overhead. In order to use concentrations only on this sequence of sets, we also need to develop a modified definition of optimism. See Appendix B and Appendix C.2 for details.

**Connection between Autonomous Exploration and Multi-Goal SSP.** In standard RL setting, it is known that in order to obtain a tight dependency on $L$, one needs to exploit the variance information in the value function (Azar et al., 2017). However, in autonomous exploration, it is unclear how to exploit the variance information because even which state is in $\mathcal{S}^{\rightarrow}_L$ is unknown.

To this end, we first consider a simpler problem, multi-goal SSP, and extend the technique for single-goal SSP (Tarbouriech et al., 2021) to this new problem (cf. Alg. 3). We also present a reduction from autonomous exploration to multi-goal SSP (cf. Alg. 2). These two techniques together yield the first tight dependency on $L$ for autonomous exploration.

## 2 Preliminaries

In this section, we introduce basic definitions and our problem setup.

**Notations.** For any two vectors $X, Y \in \mathbb{R}^S$, we write their inner product as $XY := \sum_{s \in \mathcal{S}} X(s)Y(s)$. We denote $\|X\|_\infty := \max_{s \in \mathcal{S}} |X(S)|$, and if $X$ is a probability distribution on $\mathcal{S}$, we define $\mathbb{V}(X, Y) := \sum_{s \in \mathcal{S}} X(s)Y(s)^2 - (\sum_{s \in \mathcal{S}} X(s)Y(s))^2$.

**Markov Decision Process.** We consider an MDP $M := \langle \mathcal{S}, \mathcal{A}, P, c, s_0 \rangle$, where $\mathcal{S}$ is the state space with size $S$, $\mathcal{A}$ is the action space with size $A$, and $s_0 \in \mathcal{S}$ is the initial state. In state $s$, taking action $a$ has a cost drawn i.i.d. from a distribution on $[c_{\min}, 1]$ (where $c_{\min} > 0$) with expectation $c(s, a)$, and transits to the next state $s'$ with probability $P(s'|s, a)$. For convenience, we use $P_{s,a}$ and $P_{s,a,s'}$ to denote $P(\cdot|s, a)$ and $P(s'|s, a)$, respectively. A deterministic and stationary policy $\pi : \mathcal{S} \to \mathcal{A}$ is a mapping, and the agent following the policy $\pi$ will take action $\pi(s)$ at state $s$.

For a fixed state $g \in \mathcal{S}$ we define the random variable $t^\pi_g(s)$ as the number of steps it takes to reach state $g$ starting from state $s$ when executing policy $\pi$, i.e. $t^\pi_g(s) := \inf\{t \geq 0 : s_{t+1} = g \mid s_1 = s, \pi\}$. A policy $\pi$ is a proper policy if for any state $s \in \mathcal{S}$, $t^\pi_g(s) < +\infty$ with probability 1. Then we define the value function of a proper policy $\pi$ with respect to the goal state $g$ and its corresponding $Q$-function as follows:

$$V^\pi_g(s) = \mathbb{E}\left[\sum_{t=1}^{t^\pi_g(s)} c_t(s_t, \pi(s_t)) \mid s_1 = s\right], Q^\pi_g(s, a) = \mathbb{E}\left[\sum_{t=1}^{t^\pi_g(s)} c_t(s_t, \pi(s_t)) \mid s_1 = s, \pi(s_1) = a\right],$$

where $c_t \in [c_{\min}, 1]$ is the instantaneous cost at step $t$ incurred by the state-action pair $(s_t, \pi(s_t))$, and the expectation is taken over the random sequence of states generated by executing $\pi$ starting from state $s \in \mathcal{S}$. Here we have $V^\pi_g(g) = 0$. We use $\pi_Q$ to denote the greedy policy over a vector $Q \in \mathbb{R}^{S \times A}$, i.e. $\pi_Q(s) := \arg\min_{a \in \mathcal{A}} Q(s, a)$.

For a fixed state $g \in \mathcal{S}$, we denote $V^*_g$ as the value function of the optimal policy with respect to goal state $g$, and here we list some important properties of $V^*_g$: there exists a stationary, deterministic and proper policy $\pi^*$, such that its value function $V^*_g := V^{\pi^*}_g$ and its corresponding $Q$-function $Q^*_g := Q^{\pi^*}_g$ satisfies the following Bellman optimality equations (cf. Lem. 1):

$$Q^*_g(s, a) = c(s, a) + P_{s,a} V^*_g, \qquad V^*_g(s) = \min_{a \in \mathcal{A}} Q^*_g(s, a), \qquad \forall (s, a) \in \mathcal{S} \times \mathcal{A}.$$

**Autonomous Exploration.** Now we introduce the Autonomous Exploration problem. To formally discuss the setting, we need the following assumption on our MDP $M$.

**Assumption 1.** *The action space contains a* RESET *action s.t.* $P(s_0|s, \text{RESET}) = 1$ *for any* $s \in \mathcal{S}$. *Moreover, taking* RESET *in any state* $s$ *will incur a cost* $c_{\text{RESET}}$ *with probability* 1, *where* $c_{\text{RESET}}$ *is a constant in* $[c_{\min}, 1]$.

Given any fixed length $L \geq 1$, the agent needs to learn the set of incrementally controllable states $\mathcal{S}_L^{\rightarrow}$. To introduce the concept of $\mathcal{S}_L^{\rightarrow}$, we first give the definition of policies restricted on a subset:

**Definition 1** (Policy restricted on a subset). *For any* $\mathcal{S}' \subseteq \mathcal{S}$, *a policy* $\pi$ *is restricted on the set* $\mathcal{S}'$ *if* $\pi(s) = \text{RESET}$ *for all* $s \notin \mathcal{S}'$.

Now we discuss the optimal policy restricted on a set of states $\mathcal{K} \subseteq \mathcal{S}$ with respect to goal state $g$. We denote $V_{\mathcal{K},g}^* \in \mathbb{R}^S$ as the value function of the optimal policy restricted on $\mathcal{K}$ with goal $g$, and $Q_{\mathcal{K},g}^*$ as the $Q$-function corresponding to $V_{\mathcal{K},g}^*$. We consider the case that there exists at least one proper policy restricted on $\mathcal{K}$ with the goal state $g$. Then, $V_{\mathcal{K},g}^*$ and $Q_{\mathcal{K},g}^*$ are finite, and they satisfy the following Bellman equations:

$$
\begin{aligned}
Q_{\mathcal{K},g}^*(s,a) &= c(s,a) + P_{s,a}V_{\mathcal{K},g}^*, & \forall(s,a) \in \mathcal{S} \times \mathcal{A}, \\
V_{\mathcal{K},g}^*(s) &= \min_{a \in \mathcal{A}} Q_{\mathcal{K},g}^*(s,a), & \forall s \in \mathcal{K}, s \neq g, \\
V_{\mathcal{K},g}^*(s) &= Q_{\mathcal{K},g}^*(s, \text{RESET}) = c_{\text{RESET}} + V_{\mathcal{K},g}^*(s_0), & \forall s \notin \mathcal{K} \cup \{g\}, \\
V_{\mathcal{K},g}^*(g) &= 0.
\end{aligned}
$$

We note that when $\mathcal{K}_1 \subseteq \mathcal{K}_2$, for any $g \in \mathcal{S}$, if $V_{\mathcal{K}_1,g}^*$ is finite, then $V_{\mathcal{K}_2,g}^*$ is also finite, and we have $V_{\mathcal{K}_2,g}^* \leq V_{\mathcal{K}_1,g}^*$ component-wise. And we note that for any $s \neq g$, we have $\min_{a \in \mathcal{A}} Q_{\mathcal{K},g}^*(s,a) \leq V_{\mathcal{K},g}^*(s)$. Now we introduce the definition of incrementally controllable states $\mathcal{S}_L^{\rightarrow}$ (see Tarbouriech et al. (2020) for more intuitions on this definition.):

**Definition 2** (Incrementally $L$-controllable states $\mathcal{S}_L^{\rightarrow}$). *Let* $\prec$ *be any partial order on* $\mathcal{S}$. *We denote* $\mathcal{S}_L^{\prec}$ *as the set of states reachable from* $s_0$ *with expected cost no more than* $L$ *w.r.t.* $\prec$, *which is defined as follows:*

- $s_0 \in \mathcal{S}_L^{\prec}$,
- *if there is a policy* $\pi$ *restricted on* $\{s' \in \mathcal{S}_L^{\prec} : s' \prec s\}$ *such that* $V_s^{\pi}(s_0) \leq L$, *then* $s \in \mathcal{S}_L^{\prec}$.

*The set of incrementally $L$-controllable states* $\mathcal{S}_L^{\rightarrow}$ *is given by* $\mathcal{S}_L^{\rightarrow} = \bigcup_{\prec} \mathcal{S}_L^{\prec}$.

**Learning Objective.** In our settings, the learning agent knows the constants $S$, $A$ and $c_{\min}$, but the learning agent has no prior knowledge of the transition probability $P$ or the cost function $c(\cdot, \cdot)$ of the MDP $M$. We fix the length $L \geq 1$ and an error parameter $\varepsilon \in (0, 1]$. A learning algorithm of the autonomous exploration problem should output a set of states $\mathcal{K} \subseteq \mathcal{S}$ that satisfies:

- $\mathcal{S}_L^{\rightarrow} \subseteq \mathcal{K}$, i.e., the algorithm discovers all the states that we want to explore.

The algorithm also outputs a set of policies $\{\pi_s\}_{s \in \mathcal{K}}$ that satisfy one of the following criteria:

1. $(\text{AX}_L \text{ on } \mathcal{S}_L^{\rightarrow}) \; \forall s \in \mathcal{S}_L^{\rightarrow}, V_s^{\pi_s}(s_0) \leq (1+\varepsilon)L$;
2. $(\text{AX}^* \text{ on } \mathcal{S}_L^{\rightarrow}) \; \forall s \in \mathcal{S}_L^{\rightarrow}, V_s^{\pi_s}(s_0) \leq V_{\mathcal{S}_L^{\rightarrow},s}^*(s_0) + \varepsilon L$;
3. $(\text{AX}_L \text{ on } \mathcal{K}) \; \forall s \in \mathcal{K}, V_s^{\pi_s}(s_0) \leq (1+\varepsilon)L$;
4. $(\text{AX}^* \text{ on } \mathcal{K}) \; \forall s \in \mathcal{K}, V_s^{\pi_s}(s_0) \leq V_{\mathcal{K},s}^*(s_0) + \varepsilon L$.

We note that $\text{AX}^*$ on $\mathcal{K}$ is stronger than both $\text{AX}_L$ on $\mathcal{S}_L^{\rightarrow}$ and $\text{AX}^*$ on $\mathcal{S}_L^{\rightarrow}$, but it is not necessarily stronger than $\text{AX}_L$ on $\mathcal{K}$, because we do not necessarily have $V_{\mathcal{K},s}^*(s_0) \leq L$ when $s \notin \mathcal{S}_L^{\rightarrow}$. In the literature, $\text{AX}_L$ on $\mathcal{S}_L^{\rightarrow}$ was studied in the original paper that proposed the autonomous exploration problem (Lim and Auer, 2012) and $\text{AX}^*$ on $\mathcal{K}$ condition was studied in (Tarbouriech et al., 2020).

We denote $T$ as the total number of steps the agent uses, and denote $(s_t, a_t)$ as the state-action pair at the $t$-th step. We denote by $c_t(s_t, a_t)$ the instantaneous cost incurred at the $t$-th step. The performance of an algorithm is measured by the cumulative cost: $C_T := \sum_{t=1}^{T} c_t(s_t, a_t)$.

**Multi-goal SSP.** We also study a new problem, multi-goal SSP, a natural generalization of the classical SSP problem. In multi-goal SSP, we consider an MDP and a fixed length $L \geq 1$. The MDP $M$ satisfies Asmp. 1, and all of its states are incrementally $L$-controllable, i.e. $\mathcal{S}_L^{\rightarrow} = \mathcal{S}$.

---

**Algorithm 1: V**alue-**O**ptimistic **I**ncremental **S**tate **D**iscovery (`VOISD`)

1  **Input:** MDP $M = \langle \mathcal{S}, \mathcal{A}, P, c, s_0 \rangle$, confidence $\delta \in (0, 1)$, error parameter $\varepsilon \in (0, 1]$, and $L \geq 1$.

2  Initialize $\mathcal{U} \leftarrow \{\}, \mathcal{K} \leftarrow \{\}$, and $s_{\text{new}} \leftarrow s_0$. Specify constants $c_1 = 6$, $c_2 = 72$, $c_3 = 2\sqrt{2}$, $c_4 = 2\sqrt{2}$.

3  Set $\varepsilon \leftarrow \varepsilon/3$, $\delta \leftarrow \delta/2$, $B \leftarrow 10L$ and $\pi_{s_0} \in \Pi(\{s_0\})$.

4  Set $\psi \leftarrow 12000(\frac{L}{c_{\min}\varepsilon})^2 \ln(\frac{SA}{\delta})$, and $\phi \leftarrow 2^{\lceil \log_2 \psi \rceil}$.

5  For $(s, a, s') \in \mathcal{S} \times \mathcal{A} \times \mathcal{S}$, set

    $N(s, a) \leftarrow 0$; $n(s, a) \leftarrow 0$; $N(s, a, s') \leftarrow 0$; $\widehat{P}_{s,a,s'} \leftarrow 0$; $\theta(s, a) \leftarrow 0$; $\widehat{c}(s, a) \leftarrow 0$.

6  **for** round $r = 1, 2, \cdots$ **do**

7     |  \\\(a) *Discover Possible States in* $\mathcal{S}_L^{\rightarrow}$

8     |  Add $s_{\text{new}}$ to $\mathcal{K}$. Set $s \leftarrow s_{\text{new}}$.

9     |  **for** each $a \in \mathcal{A}$ **do**

10     |    |  **while** $N(s, a) < \phi$ **do**

11     |    |    |  Execute policy $\pi_s$ on MDP $M$ until reaching state $s$.

12     |    |    |  Take action $a$, incur cost $c$ and observe next state $s' \sim P(\cdot \mid s, a)$.

13     |    |    |  Set $N(s, a) \leftarrow N(s, a) + 1$, $\theta(s, a) \leftarrow \theta(s, a) + c$, $N(s, a, s') \leftarrow N(s, a, s') + 1$.

14     |    |    |  If $s' \notin \mathcal{K}$, add $s'$ to $\mathcal{U}$.

15     |    |  Set $\widehat{c}(s, a) \leftarrow \frac{\theta(s,a)}{N(s,a)}$ and $\theta(s, a) \leftarrow 0$.

16     |    |  For all $s' \in \mathcal{S}$, set $\widehat{P}_{s,a,s'} \leftarrow N(s, a, s')/N(s, a)$, $n(s, a) \leftarrow N(s, a)$.

17     |  Stop the algorithm if $\mathcal{U}$ is empty.

18     |  \\\(b) *Compute Optimistic Policy*

19     |  For each $g \in \mathcal{U}$, compute $(Q_g, V_g) := \text{VISGO}(\mathcal{S}, \mathcal{A}, \mathcal{K}, s_0, g, \frac{c_{\min}\varepsilon}{18})$.

20     |  Choose a state $s \in \arg\min_{g \in \mathcal{U}} V_g(s_0)$. Stop the algorithm if $V_s(s_0) > L$.

21     |  Set the policy $\tilde{\pi}$ as the greedy policy over $Q_s$. Remove $s$ from $\mathcal{U}$, set $s_{\text{new}} \leftarrow s$ and set $\pi_s \leftarrow \tilde{\pi}$.

22  **Output:** The discovered states $\mathcal{K}$ and their corresponding policies $\{\pi_s\}_{s \in \mathcal{K}}$.

---

In multi-goal SSP, a learning algorithm should output a set of policies $\{\pi_s\}_{s \in \mathcal{S}}$, such that $V_s^{\pi_s}(s_0) \leq V_s^*(s_0) + \varepsilon L$ for all $s \in \mathcal{S}$. We observe that an algorithm that solves the autonomous exploration problem with AX$^*$ on $\mathcal{S}_L^{\rightarrow}$ criterion can also solve the multi-goal SSP problem.

## 3  Algorithms and Sample Complexity Bounds

Now we are ready to describe our algorithms. There are three key components. The first component aims to discover a set of states $\mathcal{K}$ which is a superset of $\mathcal{S}_L^{\rightarrow}$ but it is also not too much larger than $\mathcal{S}_L^{\rightarrow}$ (cf. Alg. 1), and compute a set of policies $\{\pi_s\}_{s \in \mathcal{K}}$ to reach each state with a small cost. The second component reduces the autonomous exploration problem to multi-goal SSP using the set $\mathcal{K}$ computed from the first component (cf. Alg. 2). Then it collects fresh samples for all state-action pairs and computes a set of near-optimal policies $\{\pi_s\}_{s \in \mathcal{K}}$ on the set $\mathcal{K}$. In the third component, inspired by recent advances in stochastic shortest path Tarbouriech et al. (2021), we design a policy evaluation step to obtain near-optimal estimates of the costs of getting to each $s \in \mathcal{S}_L^{\rightarrow}$ (cf. Alg. 3).

### 3.1  State Discovery via a Value-Optimistic Method

In Alg. 1, we compute a set of states $\mathcal{K}$ that contains $\mathcal{S}_L^{\rightarrow}$. The algorithm maintains a set of "known" states $\mathcal{K}$ and a set of "unknown" states $\mathcal{U}$ that have the potential to be recognized as members of $\mathcal{S}_L^{\rightarrow}$ and become "known" states in future. Initially, the set of known states is empty, and $s_0$ is the first state that will be added to $\mathcal{K}$. Our algorithm will discover all the states $s \in \mathcal{S}_L^{\rightarrow}$ one by one, and when the algorithm finds a policy that can reach $s$ from $s_0$ with expected cost less than $L$, the algorithm adds $s$ to the set of known states $\mathcal{K}$. Now we discuss this process in detail.

In each round, we have two phases. The first phase is called state discovery phase, and it borrows the idea from (Tarbouriech et al., 2020) and (Lim and Auer, 2012). We add a new state $s$ to the set of known states $\mathcal{K}$. Then for each action $a$, we sample $\phi$ times, compute the empirical probability $\widehat{P}_{s,a,s'}$, and use the empirical $\widehat{c}(s, a)$. The collection of these samples has two objectives. First, it will discover new states because it adds to $\mathcal{U}$ all the states that have not been discovered. Second, the collection of these samples helps us estimate the transition model $P_{s,a,s'}$, which is crucial in estimating the value function of optimal policies.

---

**Algorithm 2:** **Re**duce Autonomous Exploration to **Multi-Goal SSP** (Re-MG-SSP)

---

1 **Input:** MDP $M = \langle \mathcal{S}, \mathcal{A}, P, c, s_0 \rangle$, confidence $\delta \in (0, 1)$, error parameter $\varepsilon \in (0, 1]$, and $L \geq 1$.
2 **Input:** The states $s$ in $\mathcal{K}$ and their corresponding policy $\pi_s$.
3 For $(s, a, s') \in \mathcal{S} \times \mathcal{A} \times \mathcal{S}$, set
$\quad N(s, a) \leftarrow 0; \; n(s, a) \leftarrow 0; \; N(s, a, s') \leftarrow 0; \; \widehat{P}_{s,a,s'} \leftarrow 0; \; \theta(s, a) \leftarrow 0; \; \widehat{c}(s, a) \leftarrow 0.$
4 Set $\varepsilon \leftarrow \varepsilon/3, \delta \leftarrow \delta/2, \psi \leftarrow 12000(\frac{L}{c_{\min}\varepsilon})^2 \ln(\frac{SA}{\delta})$, and $\phi \leftarrow 2^{\lceil \log_2 \psi \rceil}$.
5 Set $\mathcal{K}^\dagger = \mathcal{K} \cup \{x\}$ where $x$ is defined in and construct an MDP $M^\dagger = \langle \mathcal{K}^\dagger, \mathcal{A}, P^\dagger, c^\dagger, s_0 \rangle$ where $x, \mathcal{K}^\dagger$,
$\quad P^\dagger, c^\dagger$ are defined in Sect. 3.2.
6 **for** each $(s, a) \in \mathcal{K} \times \mathcal{A}$ **do**
7 $\quad$ **while** $N(s, a) < \phi$ **do**
8 $\quad\quad$ Execute policy $\pi_s$ on MDP $M^\dagger$ until reaching state $s$.
9 $\quad\quad$ Take action $a$, incur cost $c$ and observe next state $s' \sim P^\dagger(\cdot \mid s, a)$.
10 $\quad\quad$ Set $N(s, a) \leftarrow N(s, a) + 1, \theta(s, a) \leftarrow \theta(s, a) + c, N(s, a, s') \leftarrow N(s, a, s') + 1.$
11 $\quad$ Set $\widehat{c}(s, a) \leftarrow \frac{\theta(s,a)}{N(s,a)}$ and $\theta(s, a) \leftarrow 0$.
12 $\quad$ For all $s' \in \mathcal{K}^\dagger$, set $\widehat{P}_{s,a,s'} \leftarrow N(s, a, s')/N(s, a), \; n(s, a) \leftarrow N(s, a).$
13 For all $a \in \mathcal{A}$, set $N(x, a) = n(x, a) \leftarrow \phi, \widehat{c}(x, a) \leftarrow c_{\text{RESET}}, \widehat{P}_{x,a,s_0} \leftarrow 1, \widehat{P}_{x,a,s'} \leftarrow 0$ for all $s' \in \mathcal{S}.$
14 For $g \in \mathcal{K}$, compute $(Q_g, V_g) := \text{VISGO}(\mathcal{K}^\dagger, \mathcal{A}, \mathcal{K}^\dagger, s_0, g, \frac{c_{\min}\varepsilon}{18})$, and set $\pi_g$ as the greedy policy over $Q_g$.
15 **Output:** $\mathcal{K}, \{\pi_s\}_{s \in \mathcal{K}}, N(), n(), \widehat{P}, \theta(), \widehat{c}, M^\dagger.$

---

In the second phase, we use the estimated model $\widehat{P}$ to estimate the value function of the optimal policy with respect to all the goal states $g \in \mathcal{U}$, and we denote the estimated value as $V_g$. We can prove that $V_g$ satisfies the optimism property. Hence, if $g \in \mathcal{S}_L^\rightarrow$, $g$ will be eventually added to the set of known states $\mathcal{K}$, and we can obtain $\mathcal{K} \supseteq \mathcal{S}_L^\rightarrow$ when the algorithm terminates. To this purpose, we use a procedure called VISGO (Value Iteration with Slight Goal Optimism), which is a modified version of the VISGO procedure in (Tarbouriech et al., 2021).

In our VISGO procedure, we use slightly different bonus function $b(U, s, a)$. The VISGO procedure in (Tarbouriech et al., 2021) computes the optimistic policy on the whole set $\mathcal{S}$, and here we make VISGO estimate the optimistic policy restricted on the set $\mathcal{K}$. As we have collected $\phi = \widetilde{O}(L^2/(c_{\min}\varepsilon)^2)$ samples for each state-action pair $(s, a) \in \mathcal{K} \times \mathcal{A}$, we can ensure that the estimated model $\widehat{P}$ is close enough to $P$, and in VISGO procedure, the bonus function $b(U, s, a)$ is bounded by $O(c_{\min}\varepsilon)$, and this is essential for our proof of $\mathcal{K} \subseteq \mathcal{S}_{(1+\varepsilon)L}^\rightarrow$. The full algorithm of VISGO is deferred to Appendix C. Now we present our upper bound of the Alg.1.

**Theorem 1.** *Assume that $L \geq 1$, $0 < \varepsilon \leq 1$ and $0 < \delta < 1$. For any MDP $M = \langle \mathcal{S}, \mathcal{A}, P, c, s_0 \rangle$ satisfying Asmp. 1, with probability at least $1 - \delta$, our Alg. 1 will terminate and output a set of states $\mathcal{K}$ such that $\mathcal{S}_L^\rightarrow \subseteq \mathcal{K} \subseteq \mathcal{S}_{(1+\varepsilon)L}^\rightarrow$, and a set of policies $\{\pi_s\}_{s \in \mathcal{K}}$ such that $\forall s \in \mathcal{K}, V_s^{\pi_s}(s_0) \leq (1+\varepsilon)L$, and the cumulative cost $C_T$ satisfies $C_T = \widetilde{O}(\frac{L^3 S_{(1+\varepsilon)L} A}{c_{\min}^2 \varepsilon^2})$.*

Thm. 1 shows that our Alg. 1 meets the $AX_L$ on $\mathcal{K}$ criterion. When $c_{\min} = 1$, which is the setting considered in prior works (Lim and Auer, 2012; Tarbouriech et al., 2020), our bound is stronger than previous ones either in $L$ or $1/\varepsilon$ (cf. Table 1). We remark that our algorithm VOISD follows the same template as the DisCo algorithm in (Tarbouriech et al., 2020), which also consists of a state-discovery phase and a compute-optimistic-policy phase. However, our sample complexity saves an $S_{(1+\varepsilon)L}$ factor compared with (Tarbouriech et al., 2020), and the main reasons lie in the difference of the methods of computing optimistic policies. The DisCo algorithm applies a model-optimistic method, which will incur an additional $S_{(1+\varepsilon)L}$ factor in the bonus term $b_{s,a}$. Our algorithm VOISD is based on the value-optimistic subroutine VISGO. It uses the concentration of $(\widehat{P}_{s,a} - P_{s,a})V_{\mathcal{K}_z,g}^*$ on a series of sets $\mathcal{K}_z$ and saves an $S_{(1+\varepsilon)L}$ term on the sample complexity.

## 3.2 CONNECTION BETWEEN AUTONOMOUS EXPLORATION AND MULTI-GOAL SSP

In Alg. 2, after we compute a set of known states $\mathcal{K} \supseteq \mathcal{S}_L^\rightarrow$, we have discovered all the states that we want to explore. Alg. 2 aims to compute a set of policies $\{\pi_s\}_{s \in \mathcal{K}}$ that meets the $AX^*$ on $\mathcal{K}$ criterion. We will fix our set of known states $\mathcal{K}$, and focus only on the policies restricted on $\mathcal{K}$. Therefore, for all the states $s \notin \mathcal{K}$, we can regard them as one artificial state $x$, and the only action at state $x$ is

---

**Algorithm 3: Val**ue-**A**ware **A**utonomous **E**xploration (`VALAE`)

---

1 **Input:** MDP $M = \langle \mathcal{S}, \mathcal{A}, P, c, s_0 \rangle$, confidence $\delta \in (0, 1)$, error parameter $\varepsilon \in (0, 1]$, and $L \geq 1$.

2 **Specify:** Trigger set $\mathcal{N} \leftarrow \{2^{j-1} \ : \ j = 1, 2, \ldots\}$. Constants $c_1 = 6$, $c_2 = 72$, $c_3 = 2\sqrt{2}$, $c_4 = 2\sqrt{2}$.

3 Set $\delta \leftarrow O(\delta\varepsilon/SAL)$.

4 $\backslash\backslash$*We run Alg. 1 with $\varepsilon = 1$ and get a set $\mathcal{K}$ such that $\mathcal{S}_L^{\rightarrow} \subseteq \mathcal{K} \subseteq \mathcal{S}_{2L}^{\rightarrow}$.*

5 Run Alg. 1 with input $(M, \delta/2, 1, L)$ and we get a set $\mathcal{K}$ and a set of policies $\{\pi_s\}_{s \in \mathcal{K}}$.

6 Run Alg. 2 with input $(M, \delta/2, 1, L, \mathcal{K}, \{\pi_s\}_{s \in \mathcal{K}})$, and we obtain the MDP $M^{\dagger} = \langle \mathcal{K}^{\dagger}, \mathcal{A}, P^{\dagger}, c^{\dagger}, s_0 \rangle$ and the variables $N(), n(), \widehat{P}, \theta(), \widehat{c}$.

7 Set initial time step $t \leftarrow 1$ and trigger index $j \leftarrow 5 + \log_2 \frac{1}{c_{\min}}$.

8 Set $\epsilon \leftarrow \frac{\varepsilon}{3}$, $B \leftarrow 10L$, $\lambda = \lceil \frac{2048}{\epsilon^2} \ln^2(\frac{256}{\epsilon}) \ln(\frac{2S}{\delta}) \rceil$, and initialize $\mathcal{U} \leftarrow \mathcal{K}$, and $g \leftarrow s_0$.

9 **for** round $r = 1, 2, \cdots$ **do**

10     $\backslash\backslash$*(a) Compute Optimistic Policy*

11     Compute $(Q, V) := \mathtt{VISGO}(\mathcal{K}^{\dagger}, \mathcal{A}, \mathcal{K}^{\dagger}, s_0, g, 2^{-j}/(|\mathcal{K}^{\dagger}|A))$.

12     Set the policy $\tilde{\pi}$ as the greedy policy over $Q$.

13     $\backslash\backslash$*(b) Policy Evaluation*

14     Set $\hat{\tau} \leftarrow 0$.

15     **for** episode $k = 1, 2, \cdots, \lambda$ **do**

16         Set $s_t \leftarrow s_0$ and reset to the initial state $s_0$. Set $\hat{\tau}_k \to 0$.

17         **while** $s_t \neq g$ **do**

18             Take action $a_t = \arg\min_{a \in \mathcal{A}} Q(s_t, a)$ on MDP $M^{\dagger}$, incur cost $c_t$ and observe next state $s_{t+1} \sim P^{\dagger}(\cdot \mid s_t, a_t)$.

19             Set $(s, a, s', c) \leftarrow (s_t, a_t, s_{t+1}, c_t)$ and $t \leftarrow t + 1$.

20             Set $N(s, a) \leftarrow N(s, a) + 1$, $\theta(s, a) \leftarrow \theta(s, a) + c$, $N(s, a, s') \leftarrow N(s, a, s') + 1$.

21             **if** $N(s, a) \in \mathcal{N}$ **then**

22                 Set $j \leftarrow j + 1$. Set $\widehat{c}(s, a) \leftarrow \frac{2\theta(s,a)}{N(s,a)}$ and $\theta(s, a) \leftarrow 0$.

23                 For all $s' \in \mathcal{K}^{\dagger}$, set $\widehat{P}_{s,a,s'} \leftarrow N(s, a, s')/N(s, a)$, $n(s, a) \leftarrow N(s, a)$.

24                 Return to line 9, start a new round (the current round has been a *skipped round*).

25             Set $\hat{\tau} \leftarrow \hat{\tau} + \frac{c}{\lambda}$, $\hat{\tau}_k \leftarrow \hat{\tau}_k + c$.

26         **if** $\hat{\tau} > V(s_0) + \epsilon L$ **then**

27             Return to line 9, start a new round (the current round has been a *failure round*).

28     Set $\pi_g \leftarrow \tilde{\pi}$. Remove $g$ from $\mathcal{U}$. The current round has been a *success round*.

29     Choose another state $g \in \mathcal{U}$. Stop the algorithm if $\mathcal{U}$ is empty.

30 **Output:** The states $s$ in $\mathcal{K}$ and their corresponding policy $\pi_s$.

---

RESET. To this purpose, we will construct an MDP $M^{\dagger} := \langle \mathcal{K}^{\dagger}, \mathcal{A}, P^{\dagger}, c^{\dagger}, s_0 \rangle$ where we first define the artificial state $x$, and we set $\mathcal{K}^{\dagger} = \mathcal{K} \cup \{x\}$, and $K' = |\mathcal{K}^{\dagger}|$. For any $(s, a) \in \mathcal{K} \times \mathcal{A}$, we define $P^{\dagger}_{s,a,s'}$ as follows:

$$P^{\dagger}_{s,a,s'} = P_{s,a,s'}, \ \forall s' \in \mathcal{K}, \text{ and } P^{\dagger}_{s,a,x} = \sum_{s' \notin \mathcal{K}} P_{s,a,s'}.$$

We also define $P^{\dagger}_{x,a,s'} = \mathbb{I}[s' = s_0]$ for any $a \in \mathcal{A}$, $s' \in \mathcal{K} \cup \{x\}$. Finally, we define $c^{\dagger}(s, a) = c(s, a)$ for all $(s, a) \in \mathcal{S} \times \mathcal{A}$, and $c^{\dagger}(x, a) = c_{\text{RESET}}$ for all $a \in \mathcal{A}$. In this way, the problem reduces to solving a multi-goal SSP problem on $M^{\dagger}$ with the set of states being $\mathcal{K}^{\dagger}$.

Next, we collect $\phi$ fresh samples for each state-action pair $(s, a)$ again, and we will use these samples to compute policies $\{\pi_s\}_{s \in \mathcal{K}}$ that satisfy the AX* on $\mathcal{K}$ criterion. We remark that using fresh samples is essential for Alg. 2 to ensure these samples are independent of $\mathcal{K}$. The following theorem states the sample complexity for Alg. 2.

**Theorem 2.** *Assume that $L \geq 1$, $0 < \varepsilon \leq 1$ and $0 < \delta < 1$. For any MDP $M = \langle \mathcal{S}, \mathcal{A}, P, c, s_0 \rangle$ satisfying Asmp. 1, running Alg. 1 and Alg. 2 successively will output a set of states $\mathcal{K}$ such that with probability at least $1 - \delta$, $\mathcal{S}_L^{\rightarrow} \subseteq \mathcal{K} \subseteq \mathcal{S}_{(1+\varepsilon)L}^{\rightarrow}$, and a set of policies $\{\pi_s\}_{s \in \mathcal{K}}$ such that*

$$\forall s \in \mathcal{K}, V_s^{\pi_s}(s_0) \leq (1+\varepsilon)L \text{ and } V_s^{\pi_s}(s_0) \leq V_{\mathcal{K},s}^*(s_0) + \varepsilon L, \text{ and } C_T \text{ satisfies } C_T = \widetilde{O}\left(\frac{L^3 S_{(1+\varepsilon)L} A}{c_{\min}^2 \varepsilon^2}\right).$$

### 3.3 VALUE-AWARE ALGORITHMS FOR AUTONOMOUS EXPLORATION AND MULTI-GOAL SSP

Finally we describe our final algorithm, `Value-Aware Autonomous Exploration (VALAE)`. First, VALAE uses Alg. 1 with $\varepsilon = 1$ as a subroutine, and Alg. 1 computes a set $\mathcal{K}$ such that $\mathcal{S}_L^{\rightarrow} \subseteq \mathcal{K}$. Then we discard all the samples collected in Alg. 1, in order to ensure the independence of $\mathcal{K}$ and $\widehat{P}_{s,a}$. Second, we use Alg. 2 as a burn-in step to collect $\widetilde{\Omega}(L^2/c_{\min}^2)$ samples for each of the state-action pair $(s, a)$ so that we can use Lem. 15 to bound the expected cost of the greedy policy $\tilde{\pi}$ in the policy-evaluation phase, and use Lem. 16 to show that we can use the average cost in $\widetilde{O}(1/\epsilon^2)$ episodes to estimate the expected cost of the policy $\tilde{\pi}$.

From now on we work on the MDP $M^{\dagger}$, and we will solve the multi-goal SSP problem on $M^{\dagger}$ and compute near-optimal policies $\pi_g$ for all the goal states $g \in \mathcal{K}$. We choose the goal state $g \in \mathcal{K}$ one by one, and we move to another goal state $g$ if the average performance of the policy $\pi_g$ is close to our estimation of the optimal policy. In each round, we have two phases. In the first phase, we use VISGO to estimate the value function of the optimal policy with goal state $g$, and we set the policy $\tilde{\pi}$ as the greedy policy over $Q$. Since we do not know whether the policy $\tilde{\pi}$ is close enough to the optimal policy, in the second phase, we will execute $\tilde{\pi}$ for $\lambda = \widetilde{O}(1/\epsilon^2)$ times and check whether the average performance is close enough to our estimation of the optimal policy. In this process, we also collect samples, and use them to help us estimate the value function of the optimal policy.

In the second phase, the current round will be classified into three cases: failure round, skipped round, and success round. This borrows the idea from (Lim and Auer, 2012). If the average performance of the policy $\tilde{\pi}$ is too bad, we will consider the current round as a failure round. If the number of samples $N(s, a)$ meets the trigger set (i.e. is a power of 2), we will consider the current round as a skipped round, following the idea in (Jaksch et al., 2010). Otherwise, the current round is a success round. In the case of a failure round or a skipped round, we will not change the goal state $g$, and in the next round, we compute a new policy by VISGO using the samples collected in this round. In the case of a success round, as the average performance of the policy $\tilde{\pi}$ is close to optimal, we can set the $\tilde{\pi}$ as the policy $\pi_g$ for the goal state $g$, and choose another goal state $g$.

**Theorem 3.** *Assume that $L \geq 1$, $0 < \varepsilon \leq 1$ and $0 < \delta < 1$. For any MDP $M = \langle \mathcal{S}, \mathcal{A}, P, c, s_0 \rangle$ satisfying Asmp. 1, we have that with probability at least $1 - \delta$, our Alg. 3 will terminate and output a set of states $\mathcal{K}$ such that $\mathcal{S}_L^{\rightarrow} \subseteq \mathcal{K} \subseteq \mathcal{S}_{2L}^{\rightarrow}$, and a set of policies $\{\pi_s\}_{s \in \mathcal{K}}$ such that $\forall s \in \mathcal{K}, V_s^{\pi_s}(s_0) \leq V_{\mathcal{K},s}^*(s_0) + \varepsilon L$, and the cumulative cost $C_T$ satisfies $C_T = \widetilde{O}\left(\frac{LS_{2L}A}{\varepsilon^2} + \frac{LS_{2L}^2 A}{\varepsilon} + \frac{L^3 S_{2L}A}{c_{\min}^2}\right)$.*

Thm.3 shows that Alg.3 satisfies the $AX^*$ on $\mathcal{K}$ criterion. Note that in Thm. 3, the dependency on $L$ is tight when $\varepsilon \to 0$, because we leverage the variance information in the policy-evaluation phase, which is necessary in RL problems generally. Alg. 1, Alg. 2 and algorithms in prior work cannot use the variance information because they collect equal number of samples on each state-action pair $(s, a)$, thus the sample collection does not use the estimated value function as the guidance.

We highlight that the leading term of $C_T$ does not have $c_{\min}$. This is because the variance fundamentally does not scale with $c_{\min}$ (cf. Lem. 17 and Lem. 18). While we discover a larger set $\mathcal{K} \subseteq \mathcal{S}_{2L}^{\rightarrow}$ compared with Thm. 1 and Thm. 2, we note that if the number of the $L$-controllable states grows polynomially with respect to $L$ (which is often implicitly considered in the literature because otherwise one may need to consider the logarithmic dependency on $|\mathcal{S}_L^{\rightarrow}|$), then we have $|\mathcal{S}_{2L}^{\rightarrow}| = O(|\mathcal{S}_L^{\rightarrow}|)$, and our complexity strictly improves the existing ones and is nearly minimax optimal.

Lastly, we note that Alg. 3 enjoys a near-optimal sample complexity for multi-goal SSP:

**Theorem 4.** *(Cumulative Cost for Multi-Goal SSP) Assume that $L \geq 1$, $0 < \varepsilon \leq 1$ and $0 < \delta < 1$. For any MDP $M = \langle \mathcal{S}, \mathcal{A}, P, c, s_0 \rangle$ satisfying Asmp. 1 and $\mathcal{S}_L^{\rightarrow} = \mathcal{S}$, with probability at least $1 - \delta$, our Alg. 3 will terminate and output a set of policies $\{\pi_s\}_{s \in \mathcal{S}}$ such that $\forall s \in \mathcal{S}, V_s^{\pi_s}(s_0) \leq V_s^*(s_0) + \varepsilon L$, and the cumulative cost $C_T$ satisfies $C_T = \widetilde{O}\left(\frac{LSA}{\varepsilon^2} + \frac{LS^2 A}{\varepsilon} + \frac{L^3 SA}{c_{\min}^2}\right)$.*

## 4 A MINIMAX LOWER BOUND FOR AUTONOMOUS EXPLORATION

Here we present our lower bound for the autonomous exploration problem. To give the lower bound formally, we introduce some basic concepts about the definition of a learning algorithm, following the definitions in Domingues et al. (2021). We define $\mathcal{I}^t = (\mathcal{S} \times \mathcal{A})^{t-1} \times \mathcal{S}$ be the set of all possible

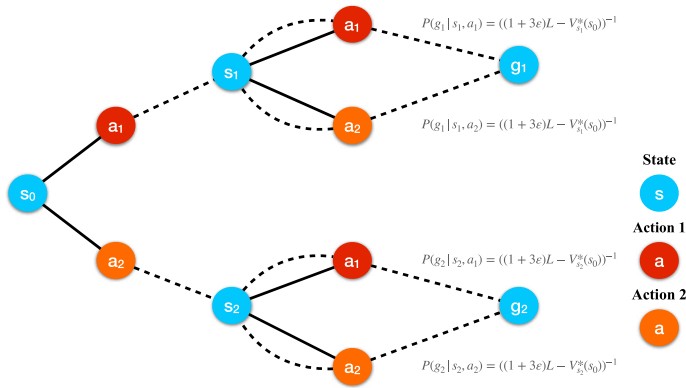

$P(g_1|s_1, a_1) = ((1 + 3\varepsilon)L - V_{s_1}^*(s_0))^{-1}$

$P(g_1|s_1, a_2) = ((1 + 3\varepsilon)L - V_{s_1}^*(s_0))^{-1}$

$P(g_2|s_2, a_1) = ((1 + 3\varepsilon)L - V_{s_2}^*(s_0))^{-1}$

$P(g_2|s_2, a_2) = ((1 + 3\varepsilon)L - V_{s_2}^*(s_0))^{-1}$

**State**

**Action 1**

**Action 2**

Figure 1: A graphical illustration of our construction of hard MDPs.

histories up to $t$ steps with the form $(s^1, a^1, s^2, a^2, \ldots, s^t) \in \mathcal{I}^t$. We denote $\boldsymbol{\pi} \triangleq (\pi^t)_{t \geq 1}$ as a history-dependent policy, where $\pi^t : \mathcal{I}^t \to \Delta(\mathcal{A})$ describes the probability of taking action $a \in \mathcal{A}$ after observing some history $i^t \in \mathcal{I}^t$ at step $t$. We consider a class of MDPs $\mathcal{M} = \langle \mathcal{S}, \mathcal{A}, P, c, s_0 \rangle$ such that Asmp. 1 holds, and we only consider the $AX_L$ on $\mathcal{S}_L^{\rightarrow}$ criterion (cf. Sect. 2). and we note that $|\mathcal{S}| = S$ and $|\mathcal{A}| = A$. Then we define an algorithm for the AX problem as a tuple $(\boldsymbol{\pi}, \tau, \mathcal{K}, \{\pi_s\}_{s \in \mathcal{K}})$, which means the algorithm executes a history-dependent policy $\boldsymbol{\pi}$, and returns a set $\mathcal{K}$ and a set of policies $\{\pi_s\}_{s \in \mathcal{K}}$ after sampling $\tau$ times.

**Definition 3.** *Given any $L \geq 1$, $0 < \varepsilon \leq 1$, $0 < \delta < 1$, and $S_L \in \mathbb{N}$, an algorithm $(\boldsymbol{\pi}, \tau, \mathcal{K}, \{\pi_s\}_{s \in \mathcal{K}})$ is $(\varepsilon, \delta)$-PAC for the AX problem within cumulative cost $C$, if for any MDP $\mathcal{M} = \langle \mathcal{S}, \mathcal{A}, P, c, s_0 \rangle$ with at most $S$ states and $A$ actions such that $\mathcal{M}$ satisfies Asmp. 1 and $|\mathcal{S}_L^{\rightarrow}| \leq S_L$, the algorithm always terminates after using $\tau$ samples, the cumulative cost is always not larger than $C$, and with probability at least $1 - \delta$, it returns a set $\mathcal{K} \supseteq \mathcal{S}_L^{\rightarrow}$ and a set of policies $\{\pi_s\}_{s \in \mathcal{K}}$, such that $\forall s \in \mathcal{S}_L^{\rightarrow}, V_s^{\pi_s}(s_0) \leq (1 + \varepsilon)L$. The cumulative cost $\mathcal{C}_{AX}(L, S, \varepsilon, \delta)$ for the AX problem is the least $C \in \mathbb{R}$ satisfying that there exists an algorithm $(\boldsymbol{\pi}, \tau, \mathcal{K}, \{\pi_s\}_{s \in \mathcal{K}})$ within cumulative cost $C$ which is $(\varepsilon, \delta)$-PAC for AX.*

The following theorem states the lower bound for the autonomous exploration problem.

**Theorem 5.** *Assume that $L > 4$, $A > 4$, $4 \leq S_L \leq \min\{(\frac{A}{2})^{\lfloor \frac{L}{2} \rfloor}, \frac{S}{2}\}$, $0 < \varepsilon < \frac{1}{4}$, and $\delta < \frac{1}{16}$. Then we have $\mathcal{C}_{AX}(L, S, \varepsilon, \delta) = \Omega(\frac{LS_L A}{\varepsilon^2} \log \frac{1}{\delta})$.*

We note that this lower bound is for the weakest criterion $AX_L$ on $\mathcal{S}_L^{\rightarrow}$, so it implies lower bounds for the other three criteria ($AX^*$ on $\mathcal{S}_L^{\rightarrow}$, $AX_L$ on $\mathcal{K}$, and $AX^*$ on $\mathcal{K}$). This lower bound further implies our upper bound (Theorem 3) is nearly minimax-optimal when $S_L$ and $S_{2L}$ are of the same order. We also have a lower bound for multi-goal SSP (cf. Appendix H).

For the proof, we first observe that even for a problem two states (an initial state $s_0$ and a goal state $g$), to determine whether $g$ is in $\mathcal{S}_L^{\rightarrow}$, one needs $\widetilde{\Omega}(LA/\varepsilon^2)$ samples. Next, we construct an MDP $\mathcal{M}_0'$ with $S_L - 1$ states and $A/2$ actions, so that each state $s$ of $\mathcal{M}_0'$ can be reached from the initial state $s_0$ with less than $L/2$ steps. Then for each state $s$ in $\mathcal{M}_0'$, we add a new state $g_s$, and each of the remaining $A/2$ actions of $s$ either transits to state $g_s$ or returns to state $s$. By setting the $p(g_s|s, a)$ to $((1 + 3\varepsilon)L - V_{\mathcal{M}_0', s}^*(s_0))^{-1}$, we can ensure that the expected cost to reach all the states $g_s$ from $s_0$ equals to $(1 + 3\varepsilon)L$. Then for each state $s$ in $\mathcal{M}_0'$, we slightly increase the probability of some action $a$ to reach $g_s$, making the expected cost to reach the state $g_s$ from $s_0$ become $L$. Under this construction, we can prove the $\widetilde{\Omega}(LS_L A/\varepsilon^2)$ lower bound. A graphical illustration is in Fig. 1 and the full proof is deferred to Appendix G.

## 5 CONCLUSION

We introduced new algorithms for the autonomous exploration problems, which improve existing ones. Along the way, we also introduced a new problem, multi-goal SSP problem, which can be of independent interest. The natural future directions include designing an algorithm with $\widetilde{O}(\frac{LS_L A}{\varepsilon^2})$ sample complexity that satisfies all the four criteria listed in Sect. 2 and improving the lower order terms in existing bounds.

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

## A    BASIC PROPERTY OF THE OPTIMAL POLICY

**Lemma 1** (Bertsekas and Tsitsiklis, 1991;Yu and Bertsekas, 2013). *Suppose that there exists a proper policy with respect to the goal state $g$ and that for every improper policy $\pi'$ there exists at least one state $s \in \mathcal{S}$ such that $V_g^{\pi'}(s) = +\infty$. Then the optimal policy $\pi^*$ is stationary, deterministic, and proper. Moreover, $V_g^* = V_g^{\pi^*}$ is the unique solution of the optimality equations $V_g^* = \mathcal{L}V_g^*$ and $V_g^*(s) < +\infty$ for any $s \in \mathcal{S}$, where for any vector $V \in \mathbb{R}^S$ the optimal Bellman operator $\mathcal{L}$ is defined as*

$$\mathcal{L}V(s) := \min_{a \in \mathcal{A}} \Big\{ c(s,a) + P_{s,a}V \Big\}. \tag{1}$$

*Furthermore, the optimal Q-value, denoted by $Q_g^* = Q_g^{\pi^*}$, is related to the optimal value function as follows*

$$Q_g^*(s,a) = c(s,a) + P_{s,a}V_g^*, \qquad V_g^*(s) = \min_{a \in \mathcal{A}} Q_g^*(s,a), \qquad \forall (s,a) \in \mathcal{S} \times \mathcal{A}. \tag{2}$$

## B    HIGH-PROBABILITY EVENT

We focus on the samples collected in Alg. 1, and we will define the high-probability event $\mathcal{G}$ to do concentration on the variables $\widehat{P}$ and $\widehat{c}$ in Alg. 1. We construct a series of sets of states $\mathcal{K}_0, \mathcal{K}_1, \cdots, \mathcal{K}_Z$ in the following way.

**Definition 4** (A series of sets $\mathcal{K}_0, \mathcal{K}_1, \cdots, \mathcal{K}_Z$). *The sets $\mathcal{K}_0, \mathcal{K}_1, \cdots, \mathcal{K}_Z$ are defined as follows:*

- *Set $\mathcal{K}_0 \leftarrow \{s_0\}$, and $Z \leftarrow 0$.*

- *(\*) Set $\mathcal{K}_{Z+1} \leftarrow \mathcal{K}_Z$. For any $s \notin \mathcal{K}_{Z+1}$, if there is a policy $\pi$ restricted on $\mathcal{K}_Z$ such that $V_s^\pi(s_0) \leq L$, then add $s$ to $\mathcal{K}_{Z+1}$.*

- *Stop if $\mathcal{K}_{Z+1} = \mathcal{K}_Z$. Otherwise set $Z \leftarrow Z+1$ and return to step (\*).*

We note that under this definition, we have $\mathcal{K}_0 \subseteq \mathcal{K}_1 \subseteq \mathcal{K}_2 \subseteq \cdots \subseteq \mathcal{K}_Z = \mathcal{S}_L^{\rightarrow}$. And each $\mathcal{K}_z$ depends only on our MDP $\mathcal{M}$, and is independent of the samples collected in our algorithm.

We define the set $\mathcal{F} := \{(\mathcal{K}_z, g) : g \in \mathcal{S}, z = 0, 1, 2, \cdots, Z, \text{ and } V_{\mathcal{K}_z,g}^*(s_0) \leq L\}$, and we note that $|\mathcal{F}| \leq S^2$. Also, we note that as we set $B = 10L$ in our algorithm, we can obtain $B_* := \max_{\mathcal{K},g \in \mathcal{F}} \|V_{\mathcal{K},g}^*\|_\infty \leq B$. Then we introduce the high-probability event $\mathcal{G}$.

**Definition 5** (High-probability event $\mathcal{G}$). *We define the event $\mathcal{G} := \mathcal{G}_1 \cap \mathcal{G}_2 \cap \mathcal{G}_3$, where*

$$\mathcal{G}_1 := \left\{ \forall (s,a) \in \mathcal{S} \times \mathcal{A}, \forall n(s,a) \geq 1 : |\widehat{c}(s,a) - c(s,a)| \leq 2\sqrt{\frac{2\widehat{c}(s,a)\iota_{s,a}}{n(s,a)}} + \frac{28\iota_{s,a}}{3n(s,a)} \right\},$$

$$\mathcal{G}_2 := \left\{ \forall (s,a,s') \in \mathcal{S} \times \mathcal{A} \times \mathcal{S}, \forall n(s,a) \geq 1 : |P_{s,a,s'} - \widehat{P}_{s,a,s'}| \leq \sqrt{\frac{2P_{s,a,s'}\iota_{s,a}}{n(s,a)}} + \frac{\iota_{s,a}}{n(s,a)} \right\},$$

$$\mathcal{G}_3 := \left\{ \forall (s,a) \in \mathcal{S} \times \mathcal{A}, \forall n(s,a) \geq 1, \forall (\mathcal{K},g) \in \mathcal{F} : |(\widehat{P}_{s,a} - P_{s,a})V_{\mathcal{K},g}^*| \leq 2\sqrt{\frac{\mathbb{V}(\widehat{P}_{s,a}, V_{\mathcal{K},g}^*)\iota_{s,a}}{n(s,a)}} + \frac{14B_*\iota_{s,a}}{3n(s,a)} \right\},$$

*where $\iota_{s,a} := 4\ln\left(\frac{24SA[n(s,a)]}{\delta}\right)$.*

**Lemma 2.** *It holds that $\mathbb{P}(\mathcal{G}) \geq 1 - \delta/4$.*

The lemma above is a direct consequence of the concentration inequalities.

**Lemma 3** (Cohen et al. (2020), Lem. B.5). *Let $\pi$ be a proper policy such that for some $d > 0$, the expected cost $V_g^\pi(s) \leq d$ for every non-goal state $s \neq g$. Then the probability that the cumulative cost of $\pi$ to reach the goal state from any state $s$ is more than $m$, is at most $2e^{-m/(4d)}$ for all $m \geq 0$.*

**Lemma 4.** *Let $\tau$ be a random variable on $[0, +\infty)$ such that $\Pr(\tau > m) \leq 2e^{-m/4d}$ for any $m \geq 0$, where $d > 0$ is a constant. We define the random variable $\hat{\tau} = \frac{1}{n} \sum_{k=1}^{n} \hat{\tau}_k$, where each $\hat{\tau}_k$ has the same distribution with $\tau$. Then for any $\epsilon > 0$, we have $\Pr(E(\tau) > \hat{\tau} + \epsilon d) \leq \exp(-\frac{n\epsilon^2}{128 \ln^2(64/\epsilon)})$.*

*Proof.* We set the constant $\Gamma = \lfloor 8d \ln(64/\epsilon) \rfloor$. Then we define the random variables $\tau_\Gamma = \min(\tau, \Gamma)$, $\check{\tau}_k = \min(\hat{\tau}_k, \Gamma)$, and $\check{\tau} = \frac{1}{n} \sum_{k=1}^{n} \check{\tau}_k$.

As each $\check{\tau}_k$ is a random variable on $[0, \Gamma]$, by Hoeffding's inequality, we have

$$\Pr(E(\tau_\Gamma) > \check{\tau} + \frac{1}{2}\epsilon d) \leq \exp(-\frac{n\epsilon^2 d^2}{2\Gamma^2}) \leq \exp(-\frac{n\epsilon^2}{128 \ln^2(64/\epsilon)}).$$

Moreover, we have

$$E(\tau) \leq E(\tau_\Gamma) + \sum_{i=1}^{\infty} i \cdot \Pr(\Gamma + i - 1 < \tau \leq \Gamma + i)$$

$$= E(\tau_\Gamma) + \sum_{m=\Gamma}^{\infty} \Pr(\tau > m)$$

$$\leq E(\tau_\Gamma) + 2 \sum_{m=\Gamma}^{\infty} \exp(-m/4d)$$

$$\leq E(\tau_\Gamma) + \frac{1}{2}\epsilon d.$$

Therefore, we obtain

$$\Pr(E(\tau) > \hat{\tau} + \epsilon d) \leq \Pr(E(\tau) > \check{\tau} + \epsilon d) \leq \Pr(E(\tau_\Gamma) > \check{\tau} + \frac{1}{2}\epsilon d) \leq \exp(-\frac{n\epsilon^2}{128 \ln^2(64/\epsilon)}).$$

$\square$

## C    ANALYSIS OF A VISGO PROCEDURE

In this section, we fix the known states $\mathcal{K}$ and the goal state $g$ and we analysis an execution of the VISGO procedure in Alg. 4. We use the value iteration of the form $V^{(i+1)} = \widetilde{\mathcal{L}}V^{(i)}$ to estimate the value funtion of the optimal policy. Here, we define the operator $\widetilde{\mathcal{L}}$ in the following way. For any $U \in \mathbb{R}^S$ such that $U \geq 0$, $U(g) = 0$, and $\|U\|_\infty \leq B$, we first define

$$\widetilde{\mathcal{L}}U(s,a) := \widehat{c}(s,a) + \widetilde{P}_{s,a}U - b(U,s,a),$$

for any $(s,a) \in \mathcal{K} \times \mathcal{A}$, where we define

$$b(U,s,a) := \max\left\{ c_1 \sqrt{\frac{\mathbb{V}(\widehat{P}_{s,a}, U)\iota_{s,a}}{n(s,a)}}, \ c_2 \frac{B\iota_{s,a}}{n(s,a)} \right\} + c_3 \sqrt{\frac{\widehat{c}(s,a)\iota_{s,a}}{n(s,a)}},$$

for any $(s,a) \in \mathcal{K} \times \mathcal{A}$, and we define the transition probability $\widetilde{P}_{s,a,s'} = \frac{n(s,a)}{n(s,a)+1}\widehat{P}_{s,a,s'} + \frac{\mathbb{I}[s'=g]}{n(s,a)+1}$ that slightly increases the probability to reach the goal $g$ at each state-action pair.

Then, we set $\widetilde{\mathcal{L}}U(s) := \min_{a \in \mathcal{A}} \widetilde{\mathcal{L}}U(s,a)$ for $s \in \mathcal{K}$ and $s \neq g$, and we set $\widetilde{\mathcal{L}}U(s) := c_{\text{RESET}} + U(s_0)$ for $s \in \mathcal{S} \setminus (\mathcal{K} \cup \{g\})$. Finally, we set $\widetilde{\mathcal{L}}U(g) := 0$.

We note that in all of Alg. 1, Alg. 2, and Alg. 3, before we executed VISGO procedure, we have collected $\phi = \widetilde{O}(L^2/(c_{\min}^2 \varepsilon^2))$ samples for each state-action pair $(s,a) \in \mathcal{K} \times \mathcal{A}$. Thus we have $n(s,a) \geq \phi$ for each $(s,a) \in \mathcal{K} \times \mathcal{A}$. Also, we have $\epsilon_{\text{VI}} \leq c_{\min}\varepsilon/18$. The lemmas of this section

---

**Algorithm 4:** Subroutine `VISGO`

---

1 **Input:** Set of states $\mathcal{S}$, set of actions $\mathcal{A}$, set of known states $\mathcal{K}$, initial state $s_0$, goal state $g$ and $\epsilon_{\mathrm{VI}}$.

2 **Global variables:** $B$, $L$, $N()$, $n()$, $\widehat{P}$, $\theta()$, $\widehat{c}()$.

3 For all $(s,a,s') \in \mathcal{K} \times \mathcal{A} \times \mathcal{S}$, set

$$\widetilde{P}_{s,a,s'} \leftarrow \frac{n(s,a)}{n(s,a)+1}\widehat{P}_{s,a,s'} + \frac{\mathbb{I}[s'=g]}{n(s,a)+1}.$$

4 For all $(s,a) \in \mathcal{K} \times \mathcal{A}$, set $\iota_{s,a} \leftarrow 4\ln\left(\frac{24SA[n(s,a)]}{\delta}\right)$.

5 Set $i \leftarrow 0$, $V^{(0)} \leftarrow 0$, $V^{(-1)} \leftarrow +\infty$.

6 **while** $\|V^{(i)} - V^{(i-1)}\|_\infty > \epsilon_{\mathrm{VI}}$ **do**

7 $\quad$ For all $(s,a) \in \mathcal{K} \times \mathcal{A}$, set

$$b^{(i+1)}(s,a) \leftarrow \max\left\{c_1\sqrt{\frac{\mathbb{V}(\widehat{P}_{s,a}, V^{(i)})\iota_{s,a}}{n(s,a)}}, c_2\frac{B\iota_{s,a}}{n(s,a)}\right\} + c_3\sqrt{\frac{\widehat{c}(s,a)\iota_{s,a}}{n(s,a)}}, \qquad (3)$$

$$Q^{(i+1)}(s,a) \leftarrow \widehat{c}(s,a) + \widetilde{P}_{s,a}V^{(i)} - b^{(i+1)}(s,a), \qquad (4)$$

$$V^{(i+1)}(s) \leftarrow \min_a Q^{(i+1)}(s,a). \qquad (5)$$

8 $\quad$ For all $s \notin \mathcal{K} \cup \{g\}$, set $V^{(i+1)}(s) \leftarrow c_{\mathrm{RESET}} + V^{(i)}(s_0)$.

9 $\quad$ Set $V^{(i+1)}(g) \leftarrow 0$, $i \leftarrow i+1$.

10 $\quad$ **if** $V^{(i)}(s_0) > L$ **then**

11 $\quad\quad$ **return** $Q^{(i)}$, $V^{(i)}$.

12 **return** $Q^{(i)}$, $V^{(i)}$.

---

are based on these conditions. We note that as $n(s,a) \geq \phi$, and by our construction of $\phi$, we have $b(U,s,a) \leq c_{\min}\varepsilon/18$ for any $(s,a) \in \mathcal{K} \times \mathcal{A}$. Therefore, if $U(g) = 0$ and $U \geq 0$ component-wise, when $n(s,a) \geq \phi$ for any $(s,a) \in \mathcal{K} \times \mathcal{A}$, we have $\widetilde{\mathcal{L}}U(s,a) \geq 0$ for any $(s,a) \in \mathcal{K} \times \mathcal{A}$.

For convenience, we define $b(U,s,a) := 0$ for any $s \notin \mathcal{K}$ and $a \in \mathcal{A}$.

**Lemma 5** (Tarbouriech et al. (2021), Lemma 12). *For any non-negative vector $U \in \mathbb{R}^S$ such that $U(g) = 0$, for any $(s,a) \in \mathcal{K} \times \mathcal{A}$, it holds that*

$$\widetilde{P}_{s,a}U \leq \widehat{P}_{s,a}U \leq \widetilde{P}_{s,a}U + \frac{\|U\|_\infty}{n(s,a)+1}.$$

The proof of the following Lem. 6 is similar with Lem. 16 in (Tarbouriech et al., 2021), but here we have two distributions $\tilde{p}$ and $p$. We give the whole proof for completeness.

**Lemma 6.** *Let $\Upsilon := \{v \in \mathbb{R}^S : v \geq 0, v(g) = 0, \|v\|_\infty \leq B\}$. Let $f : \Delta^S \times \Delta^S \times \Upsilon \times \mathbb{R} \times \mathbb{R} \times \mathbb{R} \to \mathbb{R}$ with $f(\tilde{p}, p, v, n, B, \iota) := \tilde{p}v - \max\left\{c_1\sqrt{\frac{\mathbb{V}(p,v)\iota}{n}}, c_2\frac{B\iota}{n}\right\}$, with constants $c_1 = 6$ and $c_2 \geq 2c_1^2$. Then $f$ satisfies, for all $v \in \Upsilon$, $n, \iota > 0$, $\tilde{p}, p \in \Delta^S$ s.t. $\tilde{p}(s) - \frac{1}{2}p(s) \geq 0$ for all $s \neq g$,*

    *1. $f(\tilde{p}, p, v, n, B, \iota)$ is non-decreasing in $v(s)$, i.e.*

$$\forall(v,v') \in \Upsilon^2, \ v \leq v' \implies f(\tilde{p}, p, v, n, B, \iota) \leq f(\tilde{p}, p, v', n, B, \iota);$$

    *2. $f(\tilde{p}, p, v, n, B, \iota) \leq \tilde{p}v - \frac{c_1}{2}\sqrt{\frac{\mathbb{V}(p,v)\iota}{n}} - \frac{c_2}{2}\frac{B\iota}{n} \leq \tilde{p}v - 2\sqrt{\frac{\mathbb{V}(p,v)\iota}{n}} - 14\frac{B\iota}{n}$;*

    *3. If $\tilde{p}(g) > 0$, then $f(\tilde{p}, p, v, n, B, \iota)$ is $\rho_{\tilde{p}}$-contractive in $v(s)$, with $\rho_{\tilde{p}} := 1 - p(g) < 1$, i.e.*

$$\forall(v,v') \in \Upsilon^2, \ |f(\tilde{p}, p, v, n, B, \iota) - f(\tilde{p}, p, v', n, B, \iota)| \leq \rho_{\tilde{p}}\|v - v'\|_\infty.$$

*Proof.* We use the idea in Tarbouriech et al. (2021), Lemma 14 to finish the proof.

The second claim holds by $\max\{x,y\} \geq (x+y)/2, \forall x,y$, by the choices of $c_1, c_2$ and because both $\sqrt{\frac{\mathbb{V}(p,v)\iota}{n}}$ and $\frac{B\iota}{n}$ are non-negative. To verify the first and third claims, we fix all other variables but

$v(s)$ and view $f$ as a function in $v(s)$. Because the derivative of $f$ in $v(s)$ does not exist only when $c_1\sqrt{\frac{\mathbb{V}(p,v)\iota}{n}} = c_2\frac{B\iota}{n}$, where the condition has at most two solutions, it suffices to prove $\frac{\partial f}{\partial v(s)} \geq 0$ when $c_1\sqrt{\frac{\mathbb{V}(p,v)\iota}{n}} \neq c_2\frac{B\iota}{n}$. Direct computation gives that for any $s \in \mathcal{S}$ and $s \neq g$,

$$\frac{\partial f}{\partial v(s)} = \tilde{p}(s) - c_1\mathbb{I}\left[c_1\sqrt{\frac{\mathbb{V}(p,v)\iota}{n}} \geq c_2\frac{B\iota}{n}\right]\frac{p(s)(v(s)-pv)\iota}{\sqrt{n\mathbb{V}(p,v)\iota}}$$

$$\geq \min\left\{\tilde{p}(s),\ \tilde{p}(s) - \frac{c_1^2}{c_2 B}p(s)\big(v(s)-pv\big)\right\}$$

$$\overset{(i)}{\geq} \min\left\{\tilde{p}(s),\ \tilde{p}(s) - \frac{c_1^2}{c_2}p(s)\right\}$$

$$\geq 0.$$

Here (i) is by $v(s) - pv \leq v(s) \leq B$. In addition, we have

$$\sum_{s\neq g}\left|\frac{\partial f}{\partial v(s)}\right| = \sum_{s\neq g}\left[\tilde{p}(s) - c_1\mathbb{I}\left[c_1\sqrt{\frac{\mathbb{V}(p,v)\iota}{n}} \geq c_2\frac{B\iota}{n}\right]\frac{p(s)(v(s)-pv)\iota}{\sqrt{n\mathbb{V}(p,v)\iota}}\right]$$

$$= 1 - \tilde{p}(g) - c_1\mathbb{I}\left[c_1\sqrt{\frac{\mathbb{V}(p,v)\iota}{n}} \geq c_2\frac{B\iota}{n}\right]\sqrt{\frac{\iota}{n\mathbb{V}(p,v)}}\left[pv - (1-p(g))\cdot pv\right]$$

$$\leq 1 - \tilde{p}(g).$$

Therefore, we obtain that $f$ is $\rho_{\tilde{p}}$-contractive. $\qquad\square$

We note that by definition of $\widetilde{P}_{s,a}$, we have $\widetilde{P}_{s,a,s'} - \frac{1}{2}\widehat{P}_{s,a,s'} \geq 0$ for all $(s,a,s') \in \mathcal{K} \times \mathcal{A} \times \mathcal{S}$.

The following two lemmas follow the same proof with Lem.18, Lem.19 in (Tarbouriech et al., 2021), respectively.

**Lemma 7.** *The sequence $(V^{(i)})_{i\geq 0}$ is non-decreasing.*

**Lemma 8.** $\widetilde{\mathcal{L}}$ *is a $\rho$-contractive operator with modulus $\rho := 1 - \nu < 1$, where $\nu = \min\limits_{(s,a)\in\mathcal{K}\times\mathcal{A}}\widetilde{P}_{s,a,g}$.*
*Hence, the* VISGO *procedure will terminate after at most $\lceil\log(1/\epsilon_{\mathrm{VI}})/\log(1/\rho)\rceil$ iterations.*

### C.1 THE BOUNDED ERROR PROPERTY OF VISGO

Now we focus on Alg. 1. We give the following lemma of the bounded error property (Lem. 9), which indicates that the value function of the policy $\pi_s$ is close to our estimation. The proof Lem. 9 uses the techniques of Lem. 2 in (Tarbouriech et al., 2020). Our Lem. 9 focuses on a more general operator $\widetilde{\mathcal{L}}$. In our $\widetilde{\mathcal{L}}$, we involve the bonus function $b(U,s,a)$, which is not contained in (Tarbouriech et al., 2020).

**Lemma 9.** *In Alg.1, under the event $\mathcal{G}$, for any goal state $g$ that is added to $\mathcal{K}$, let $(Q,V)$ be the output $(Q_g, V_g)$ of* VISGO *in that round, and let $\pi$ be the greedy policy with respect to $Q$. Then $\pi$ is proper on the model $P_{s,a,s'}$, and for all $s \in \mathcal{S}$, we have $V_g^\pi(s) \leq (1+\varepsilon)V(s)$.*

*Proof.* We define $\widetilde{V}_g^\pi(s)$ as the value function of $\pi$ with goal state $g$ on the model $\widetilde{P}_{s,a,s'}$. We will first prove that $\widetilde{V}_g^\pi(s) \leq (1+\varepsilon/3)V(s)$, and then prove that $V_g^\pi(s) \leq (1+\varepsilon/3)\widetilde{V}_g^\pi(s)$ using the simulation lemma on the two models $\widetilde{P}_{s,a,s'}$ and $P_{s,a,s'}$. Combining them together yields $V_g^\pi(s) \leq (1+\varepsilon)V(s)$.

First we focus on model $\widetilde{P}_{s,a,s'}$. We recall that for any $s \in \mathcal{K}$ and $s \neq g$,

$$\widetilde{\mathcal{L}}u(s) := \min_{a\in\mathcal{A}}\left\{\widehat{c}(s,a) - b(u,s,a) + \widetilde{P}_{s,a}u\right\}$$

where

$$b(u, s, a) = \max\left\{c_1\sqrt{\frac{\mathbb{V}\left(\widehat{P}_{s,a}, u\right)\iota_{s,a}}{n(s,a)}}, c_2\frac{B\iota_{s,a}}{n(s,a)}\right\} + c_3\sqrt{\frac{\iota_{s,a}}{n(s,a)}},$$

and we define $b(u, s, a) = 0$ when $s \notin \mathcal{K}$.

As we set

$$\phi = O((\frac{L}{c_{\min}\varepsilon})^2 \ln(\frac{SA}{\delta})),$$

and $n(s, a) \geq \phi$, we can obtain $b(u, s, a) \leq c_{\min}\varepsilon/18$ when $\|u\|_\infty \leq B$.

In addition, under the event $\mathcal{G}_1$, when $n(s, a) \geq \phi$, we have $|\widehat{c}(s, a) - c(s, a)| \leq c_{\min}\varepsilon/18$ for all $(s, a) \in \mathcal{S} \times \mathcal{A}$.

We denote $l$ as the final iteration index of VISGO, and $V = V^{(l)}$. In VISGO, we have $V^{(i)} = \widetilde{\mathcal{L}}V^{(i-1)}$ for all $i = 1, 2, \cdots, l$. As $V^{(l-1)} \leq V^{(l)}$ component-wise, we have for any $s \in \mathcal{S}$, $V(s) \leq V^{(l-1)}(s_0) + 1 \leq L + 1$. Hence, $\|V\|_\infty \leq L + 1 \leq B$.

We set $\gamma = c_{\min}\varepsilon/6$. As $\epsilon_{\mathrm{VI}} \leq c_{\min}\varepsilon/18$, we have $b(u, s, a) + |\widehat{c}(s, a) - c(s, a)| + \epsilon_{\mathrm{VI}} \leq \gamma$ when $\|u\|_\infty \leq B$.

Given the policy $\pi$ restricted on $\mathcal{K}$, we introduce the following operators on $\mathbb{R}^S$:

$$\mathcal{L}^\pi u(s) = \widehat{c}(s, \pi(s)) - b(u, s, \pi(s)) + \widetilde{P}_{s,\pi(s)}u,$$

$$\mathcal{T}_\gamma^\pi u(s) := c(s, \pi(s)) - \gamma + \widetilde{P}_{s,\pi(s)}u.$$

We can write component-wise

$$\mathcal{T}_\gamma^\pi V \leq \mathcal{L}^\pi V - \epsilon_{\mathrm{VI}} \stackrel{(a)}{=} \widetilde{\mathcal{L}}V - \epsilon_{\mathrm{VI}} \stackrel{(b)}{\leq} V,$$

where $(a)$ uses that $\pi$ is the greedy policy with respect to $V$. To prove (b), we recall that $V = V^{(l)} = \widetilde{\mathcal{L}}V^{(l-1)}$. By contraction property of $\widetilde{\mathcal{L}}$, we have $\|\widetilde{\mathcal{L}}V - V\|_\infty \leq \|V^{(l)} - V^{(l-1)}\|_\infty$. By stopping condition of VISGO, we have $\|V^{(l)} - V^{(l-1)}\|_\infty \leq \epsilon_{\mathrm{VI}}$, thus (b) is proved. By monotonicity of the operator $\mathcal{T}_\gamma^\pi$, we have for all $m > 0$, $(\mathcal{T}_\gamma^\pi)^m V \leq V$.

As $(\mathcal{T}_\gamma^\pi)^m V$ not increases element-wise when $m$ increase, and is non-negative element wise, when $m \to \infty$, it will converge to $W_\gamma^\pi$, where $W_\gamma^\pi$ is the value function of policy $\pi$ in the model $\widetilde{P}$ with $\gamma$ subtracted to all the costs, and we have $W_\gamma^\pi \leq V$ component-wise. We define the random variable $\tilde{t}_g^\pi(s)$ as the number of steps it takes to reach $g$ starting from $s$ on model $\widetilde{P}$ when executing policy $\pi$. Thus

$$W_\gamma^\pi(s) := \mathbb{E}_{\widetilde{P}}\left[\sum_{t=1}^{\tilde{t}_g^\pi(s)} c(s_t, \pi(s_t)) - \gamma \mid s_1 = s\right] = \widetilde{V}_g^\pi(s) - \gamma\mathbb{E}_{\widetilde{P}}\left[\tilde{t}_g^\pi(s)\right].$$

Moreover, we have $c_{\min}\mathbb{E}\left[\tilde{t}_g^\pi(s)\right] \leq \widetilde{V}_g^\pi(s)$. Therefore, we get

$$\widetilde{V}_g^\pi(s) \leq \frac{W_\gamma^\pi(s)}{1 - \gamma/c_{\min}} \leq \frac{V(s)}{1 - \gamma/c_{\min}} \leq (1 + \varepsilon/3)V(s).$$

As $n(s, a) \geq \phi$, under the event $\mathcal{G}_2$, we have $\forall(s, a, s') \in \mathcal{S} \times \mathcal{A} \times \mathcal{S}$,

$$\left|P_{s,a,s'} - \widetilde{P}_{s,a,s'}\right| \leq \frac{c_{\min}\varepsilon}{6(L+1)}.$$

Thus by simulation lemma for SSP (Lemma 3 in Tarbouriech et al. (2020)), $\pi$ is proper on true model $P_{s,a,s'}$, and we have for all $s \in \mathcal{S}$, $V_g^\pi(s) \leq (1 + \varepsilon/3)\widetilde{V}_g^\pi(s)$. The proof is completed. $\qquad\square$

### C.2 OPTIMISM OF VISGO

Now we will give the optimism property. We still focus on Alg. 1. We note that there are exponentially possible number of set $\mathcal{K}$, thus we cannot include all the $\mathcal{K}$ in our high-probability event $\mathcal{G}$. As the event $\mathcal{G}_3$ includes the sets $\mathcal{K}_0, \mathcal{K}_1, \cdots, \mathcal{K}_Z$, we choose $z \in \{0 \leq z \leq Z : \mathcal{K}_z \subseteq \mathcal{K}\}$, and we will compare the output of VISGO (denoted by $V$), with the value function of the optimal policy restricted on $\mathcal{K}_z$ with goal state $g$ (denoted by $V^*_{\mathcal{K}_z,g}$). We recall that we define $B_* = \max\limits_{(\mathcal{K},g)\in\mathcal{F}} \|V^*_{\mathcal{K},g}\|_\infty$. By the definition of $\mathcal{F}$, we have $B_* \leq L + 1 \leq B$.

**Lemma 10.** *Let $z \in \{0 \leq z \leq Z : \mathcal{K}_z \subseteq \mathcal{K}\}$. Assume that there exists at least one proper policy restricted on $\mathcal{K}_z$ with the goal state $g$, and $V^*_{\mathcal{K}_z,g}(s) \leq B$ for all $s \in \mathcal{S}$. In Alg.1, under the event $\mathcal{G}$, for any output $(Q,V)$ of the VISGO procedure, it holds that*

$$Q(s,a) \leq Q^*_{\mathcal{K}_z,g}(s,a), \qquad\qquad \forall (s,a) \in \mathcal{K} \times \mathcal{A},$$
$$V(s) \leq V^*_{\mathcal{K}_z,g}(s), \qquad\qquad \forall s \in \mathcal{S}.$$

*Proof.* We prove by induction that for any inner iteration $i$ of VISGO, $Q^{(i)}(s,a) \leq Q^*_{\mathcal{K}_z,g}(s,a)$ for any $(s,a) \in \mathcal{K}\times\mathcal{A}$, and $V^{(i)}(s) \leq V^*_{\mathcal{K}_z,g}(s)$ for any $s \in \mathcal{S}$. By definition we have $Q^{(0)} = 0 \leq Q^*_{\mathcal{K}_z,g}$, and $V^{(0)} = 0 \leq V^*_{\mathcal{K}_z,g}$. Assume that the property holds for iteration $i$, then for any $(s,a) \in \mathcal{K} \times \mathcal{A}$,

$$Q^{(i+1)}(s,a) = \widehat{c}(s,a) + \widetilde{P}_{s,a}V^{(i)} - b^{(i+1)}(s,a),$$

where

$$\widehat{c}(s,a) + \widetilde{P}_{s,a}V^{(i)} - b^{(i+1)}(s,a)$$

$$= \widehat{c}(s,a) + \widetilde{P}_{s,a}V^{(i)} - \max\left\{c_1\sqrt{\frac{\mathbb{V}(\widehat{P}_{s,a}, V^{(i)})\iota_{s,a}}{n(s,a)}}, c_2\frac{B\iota_{s,a}}{n(s,a)}\right\} - c_3\sqrt{\frac{\widehat{c}(s,a)\iota_{s,a}}{n(s,a)}}$$

$$\overset{\text{(i)}}{\leq} c(s,a) + \widetilde{P}_{s,a}V^{(i)} - \max\left\{c_1\sqrt{\frac{\mathbb{V}(\widehat{P}_{s,a}, V^{(i)})\iota_{s,a}}{n(s,a)}}, c_2\frac{B\iota_{s,a}}{n(s,a)}\right\} + \frac{28\iota_{s,a}}{3n(s,a)}$$

$$= c(s,a) + f(\widetilde{P}_{s,a}, \widehat{P}_{s,a}, V^{(i)}, n(s,a), B, \iota_{s,a}) + \frac{28\iota_{s,a}}{3n(s,a)}$$

$$\overset{\text{(ii)}}{\leq} c(s,a) + f(\widetilde{P}_{s,a}, \widehat{P}_{s,a}, V^*_{\mathcal{K}_z,g}, n(s,a), B, \iota_{s,a}) + \frac{28\iota_{s,a}}{3n(s,a)}$$

$$\overset{\text{(iii)}}{\leq} c(s,a) + \widetilde{P}_{s,a}V^*_{\mathcal{K}_z,g} - 2\sqrt{\frac{\mathbb{V}(\widehat{P}_{s,a}, V^*_{\mathcal{K}_z,g})\iota_{s,a}}{n(s,a)}} - \frac{14B\iota_{s,a}}{3n(s,a)}$$

$$\overset{\text{(iv)}}{\leq} c(s,a) + \widehat{P}_{s,a}V^*_{\mathcal{K}_z,g} - 2\sqrt{\frac{\mathbb{V}(\widehat{P}_{s,a}, V^*_{\mathcal{K}_z,g})\iota_{s,a}}{n(s,a)}} - \frac{14B\iota_{s,a}}{3n(s,a)}$$

$$\overset{\text{(v)}}{\leq} \underbrace{c(s,a) + P_{s,a}V^*_{\mathcal{K}_z,g}}_{=Q^*_{\mathcal{K}_z,g}(s,a)} - (B - B_*)\frac{14\iota_{s,a}}{3n(s,a)}$$

$$\leq Q^*_{\mathcal{K}_z,g}(s,a),$$

where (i) is by definition of $\mathcal{G}_1$ and choice of $c_3$, (ii) uses the first property of Lem. 6 and the induction hypothesis that $V^{(i)} \leq V^*_{\mathcal{K}_z,g}$, (iii) uses the second property of Lem. 6 and assumption $B \geq \max\{B_*, 1\}$, (iv) uses Lem. 5, (v) is by definition of $\mathcal{G}_3$. Ultimately, for any $(s,a) \in \mathcal{K} \times \mathcal{A}$,

$$Q^{(i+1)}(s,a) \leq Q^*_{\mathcal{K}_z,g}(s,a).$$

Then for any $s \in \mathcal{K}$ and $s \neq g$, we have $V^{(i+1)}(s) = \min\limits_{a\in\mathcal{A}} Q^{(i+1)}(s,a) \leq \min\limits_{a\in\mathcal{A}} Q^*_{\mathcal{K}_z,g}(s,a) \leq V^*_{\mathcal{K}_z,g}(s)$.

In addition, $V^{(i+1)}(g) = 0 = V^*_{\mathcal{K}_z,g}(g)$, and for any $s \notin \mathcal{K} \cup \{g\}$, as $\mathcal{K}_z \subseteq \mathcal{K}$, we have

$$V^{(i+1)}(s) = c_{\text{RESET}} + V^{(i)}(s_0) \leq c_{\text{RESET}} + V^*_{\mathcal{K}_z,g}(s_0) = V^*_{\mathcal{K}_z,g}(s).$$

This completes the proof of this lemma. □

# D  PROOF OF THM. 1

*Proof idea.* We provide the main intuition here. We denote $\mathcal{K}$ and $\{\pi_s\}_{s \in \mathcal{K}}$ as the output of Alg.1. We fix the round index $r$, and we consider a state $g \notin \mathcal{K}$, $g \in \mathcal{S}_L^{\rightarrow}$ and $V_{\mathcal{K},g}^*(s_0) \leq L$. To show that $\mathcal{S}_L^{\rightarrow} \subseteq \mathcal{K}$, we need two steps. First, we show that $g$ will be added to $\mathcal{U}$ under the event $\mathcal{G}_2$. Then we will use the optimism property of VISGO (Lem. 10) to prove that our estimation of the value funtion of the optimal policy $V_g(s_0)$ is no larger than $L$.

To prove that $\mathcal{K} \subseteq \mathcal{S}_{(1+\varepsilon)L}^{\rightarrow}$, we need to prove that when $g$ is added to $\mathcal{K}$, we have $V_g^{\tilde{\pi}}(s_0) \leq (1+\varepsilon)L$, where $\tilde{\pi}$ is the greedy policy over $Q_g$ in that round. This is a direct consequence of Lem. 9.

Now we bound the total cumulative cost. For each state-action pair $(s, a) \in \mathcal{K} \times \mathcal{A}$, we collected $\widetilde{O}(L^2/(c_{\min}^2\varepsilon^2))$ samples. And to reach each $s \in \mathcal{K}$, we executed the policy $\pi_s$, and the cost to reach $s$ is no larger than $\widetilde{O}(L)$ with high probability. Thus the total cost can be bounded by $\widetilde{O}(L^3 KA/(c_{\min}^2\varepsilon^2))$, and $K = |\mathcal{K}| \leq S_{(1+\varepsilon)L}$.

$\square$

*Proof.* We first use Lem. 10 to show that $\mathcal{S}_L^{\rightarrow} \subseteq \mathcal{K}$. Then we use Lem. 9 to show that $\mathcal{K} \subseteq \mathcal{S}_{(1+\varepsilon)L}^{\rightarrow}$, and each policy $\pi_s$ output by Alg.1 satisfies $V_s^{\pi_s}(s_0) \leq (1 + \varepsilon)V_{\mathcal{K},s}^*(s_0)$. Then we will bound the total sample complexity of Alg.1.

To show that $\mathcal{S}_L^{\rightarrow} \subseteq \mathcal{K}$, we will prove that $\mathcal{K}_z \subseteq \mathcal{K}$ for all $z = 0, 1, 2, \cdots, Z$ by induction. We observe that $\{s_0\} = \mathcal{K}_0 \subseteq \mathcal{K}$ because the initial $s_{\text{new}}$ is $s_0$. Then we assume that $\mathcal{K}_z \subseteq \mathcal{K}$, and we will prove $\mathcal{K}_{z+1} \subseteq \mathcal{K}$.

For any state $g \in \mathcal{K}_{z+1}$ and $g \notin \mathcal{K}_z$, we have $V_{\mathcal{K}_z,g}^* \leq L$ by definition of $\mathcal{K}_{z+1}$. To show that $g$ will be added to $\mathcal{K}$ eventually, we need to prove the following two statements: (1) $g$ is added to $\mathcal{U}$; (2) the output of VISGO procedure $V_g(s_0) \leq L$ when $\mathcal{K}_z \subseteq \mathcal{K}$. We observe that as $V_{\mathcal{K}_z,g}^* \leq L$, there must exist $(s, a) \in \mathcal{K}_z \times \mathcal{A}$ such that $P_{s,a,g} \geq 1/L$. As we have collected $\phi = \widetilde{O}(L^2)$ samples for each of the state-action pair $(s, a) \in \mathcal{K} \times \mathcal{A}$, under event $\mathcal{G}_2$, we can obtain that $\widehat{P}_{s,a,g} \geq \frac{3}{4}L > 0$, which yields that $g$ has been discovered. Thus $g$ has been added to $\mathcal{U}$ in some round, and statement (1) is verified. The statement (2) is a direct consequence of the optimism property Lem. 10. Thus we have proved that $\mathcal{K}_{z+1} \subseteq \mathcal{K}$, which yields that $\mathcal{S}_L^{\rightarrow} \subseteq \mathcal{K}$.

Then we will prove $\mathcal{K} \subseteq \mathcal{S}_{(1+\varepsilon)L}^{\rightarrow}$. By Lem. 9, each time when we add $s$ to $\mathcal{K}$, we have $V_s^{\pi_s}(s_0) \leq (1+\varepsilon)L$, i.e. the $\text{AX}_L$ on $\mathcal{K}$ criterion is met. We recall that the definition of $\mathcal{S}_{(1+\varepsilon)L}^{\rightarrow}$ is a recursive construction, adding new state $s$ whenever there exists some policy $\pi$ such that $V_s^{\pi}(s_0) \leq (1+\varepsilon)L$. Therefore, each state added to $\mathcal{K}$ is a subset of $\mathcal{S}_{(1+\varepsilon)L}^{\rightarrow}$, and we obtain that $\mathcal{K} \subseteq \mathcal{S}_{(1+\varepsilon)L}^{\rightarrow}$.

Now we will bound the total cumulative cost. We note that we have collected $\phi = \widetilde{O}(L^2/(c_{\min}^2\varepsilon^2))$ samples for each state-action pair $(s, a) \in \mathcal{K} \times \mathcal{A}$. To collect each sample $(s, a, s', c)$, we executed a policy $\pi_s$ to reach the state $s$, and the expected cost $V_s^{\pi_s}(s_0) \leq (1 + \varepsilon)L \leq 2L$. By Lem. 3, we obtain that with probability at least $1 - \delta/4$, for any state $s$, each time when the policy $\pi_s$ is executed, the total cost to reach $s$ from $s_0$ is no larger than $O(L \log(S/\delta))$. Therefore, the total cumulative cost in Part 1 of Alg. 1 can be bounded by $\widetilde{O}(L^3 S_{(1+\varepsilon)L}A/(c_{\min}^2\varepsilon^2))$. Thus we obtain the bound

$$C_T = \widetilde{O}\left(\frac{L^3 S_{(1+\varepsilon)L}A}{c_{\min}^2\varepsilon^2}\right).$$

Now we bound the total failure probability. The event $\mathcal{G}$ fails with probability no more than $\delta/4$. And in the previous para the failure probability is bounded by $\delta/4$, thus the total failure probability is no more than $\delta/2$. The proof of Thm. 1 is completed.

$\square$

# E  PROOF OF THM. 2

First we define the high-probability event $\mathcal{E}$ to do concentration on all the samples collected in Alg. 2. We note that the definition of $\mathcal{E}$ also applies for Alg. 3 after it finishes the subroutine Alg. 1 and clears all the samples in Alg. 1. We note that after Alg.1, the set of known states $\mathcal{K}$ is fixed, and our algorithm focuses on the new MDP $\mathcal{M}^\dagger = \langle \mathcal{K}^\dagger, \mathcal{A}, P^\dagger, c^\dagger, s_0 \rangle$. Here for any $g \in \mathcal{K}$, we denote the vector $V_g^* \in \mathbb{R}^{K'}$ as the value function of the optimal policy on MDP $\mathcal{M}^\dagger$ with respect to goal $g$, and when $\mathcal{K} \subseteq \mathcal{S}_{(1+\varepsilon)L}^{\rightarrow}$ in Alg. 2, we have $B_*^\dagger := \|V_g^*\|_\infty \le B$. For convenience, here we denote $V_g^*(s)$ as the expected cost of the optimal policy to reach $g$ from $s$ on MDP $M^\dagger$, and we have $V_g^*(s) \le 2L + 1 \le B$ for all $(s, g) \in \mathcal{K}^\dagger \times \mathcal{K}$. Then we define the high-probability event $\mathcal{G}^\dagger$.

**Definition 6** (High-probability event $\mathcal{G}^\dagger$). *We define the event $\mathcal{G}^\dagger := \mathcal{G}_1^\dagger \cap \mathcal{G}_2^\dagger \cap \mathcal{G}_3^\dagger$, where*

$$\mathcal{G}_1^\dagger := \left\{ \forall (s,a) \in \mathcal{K}^\dagger \times \mathcal{A}, \forall n(s,a) \ge 1 \, : \, |\widehat{c}(s,a) - c^\dagger(s,a)| \le 2\sqrt{\frac{2\widehat{c}(s,a)\iota_{s,a}}{n(s,a)}} + \frac{28\iota_{s,a}}{3n(s,a)} \right\},$$

$$\mathcal{G}_2^\dagger := \left\{ \forall (s,a,s') \in \mathcal{K}^\dagger \times \mathcal{A} \times \mathcal{K}^\dagger, \forall n(s,a) \ge 1 \, : \, |P_{s,a,s'}^\dagger - \widehat{P}_{s,a,s'}| \le \sqrt{\frac{2P_{s,a,s'}^\dagger \iota_{s,a}}{n(s,a)}} + \frac{\iota_{s,a}}{n(s,a)} \right\},$$

$$\mathcal{G}_3^\dagger := \left\{ \forall (s,a,g) \in \mathcal{K}^\dagger \times \mathcal{A} \times \mathcal{K}, \forall n(s,a) \ge 1 \, : \, |(\widehat{P}_{s,a} - P_{s,a}^\dagger)V_g^*| \le 2\sqrt{\frac{\mathbb{V}(\widehat{P}_{s,a}, V_g^*)\iota_{s,a}}{n(s,a)}} + \frac{14B_*^\dagger \iota_{s,a}}{3n(s,a)} \right\},$$

*where $\iota_{s,a} := 4\ln\left(\frac{24SA[n(s,a)]}{\delta}\right)$.*

Finally, we define the high-probability event $\mathcal{E} := \mathcal{G} \cap \mathcal{G}^\dagger$. And we have the following lemma.

**Lemma 11.** *It holds that $\mathbb{P}(\mathcal{E}) \ge 1 - \delta/2$.*

We introduce the lemma of optimism and the lemma of bounded error in Alg. 2. We note that the proof of them is the same with Lem. 10 and Lem. 9, respectively.

**Lemma 12.** *In Alg.2, under the event $\mathcal{E}$, for any output $(Q_g, V_g)$ of the `VISGO` procedure, it holds that*

$$\begin{aligned} Q_g(s,a) &\le Q_g^*(s,a), &&\forall (s,a) \in \mathcal{K}^\dagger \times \mathcal{A}, \\ V_g(s) &\le V_g^*(s), &&\forall s \in \mathcal{K}^\dagger. \end{aligned}$$

**Lemma 13.** *In Alg. 2, under the event $\mathcal{E}$, let $(Q_g, V_g)$ be the output of `VISGO`, where $g$ is the goal state, and let $\pi$ be the greedy policy with respect to $Q_g$. Then $\pi$ is proper on the model $P_{s,a,s'}$, and for all $s \in \mathcal{K}^\dagger$, we have $V_g^\pi(s) \le (1+\varepsilon)V_g(s)$.*

Now we give the proof of Thm. 2. First we prove the correctness. By Thm. 1, `VOISD` outputs a set $\mathcal{K}$ such that $\mathcal{S}_L^{\rightarrow} \subseteq \mathcal{K} \subseteq \mathcal{S}_{(1+\varepsilon)L}^{\rightarrow}$. Now we focus on the set of policies $\{\pi_s\}_{s \in \mathcal{K}}$ that Alg. 2 outputs, and we will prove that for each $g \in \mathcal{K}$, we have $V_g^\pi(s_0) \le (1+3\varepsilon)L$ and $V_g^\pi(s_0) \le V_{\mathcal{K},g}^*(s_0) + 3\varepsilon L$. As we set $\varepsilon \leftarrow \varepsilon/3$ in the beginning of Alg. 1, the correctness of Alg. 1+Alg. 2 can be proved.

By Lem. 13 and Lem. 12, we obtain that $V_g^\pi(s_0) \le (1+\varepsilon)V_g(s_0) \le (1+\varepsilon)V_g^*(s_0)$. We note that $V_g^*(s_0) \le (1+\varepsilon)L$, because Alg. 1 satisfies $\text{AX}_L$ on $\mathcal{K}$. Hence, we can get $V_g^\pi(s_0) \le (1+\varepsilon)^2 L \le (1+3\varepsilon)L$, and $V_g^\pi(s_0) \le V_g^*(s_0) + \varepsilon(1+\varepsilon)L \le V_g^*(s_0) + 3\varepsilon L$. Thus Alg. 2 outputs a set of policies $\{\pi_s\}$ that satisfies both $\text{AX}_L$ on $\mathcal{K}$ and $\text{AX}^*$ on $\mathcal{K}$.

Similarly with Thm. 1, as we collect $\phi$ samples for each state-action pair $(s,a) \in \mathcal{K} \times \mathcal{A}$, we can obtain that with probability at least $1 - \delta$, the total sample complexity is bounded by $\widetilde{O}(\frac{L^3 S_{(1+\varepsilon)L} A}{c_{\min}^2 \varepsilon^2})$.

# F  PROOF OF THM. 3

Here we give a proof of Thm. 3 and we focus on the fixed set $\mathcal{K}^\dagger$ and our constructed MDP $M^\dagger = \langle \mathcal{K}^\dagger, \mathcal{A}, P^\dagger, c^\dagger, s_0 \rangle$. We denote $K = |\mathcal{K}|$, and $K' = |\mathcal{K}^\dagger| = K + 1$.

*Proof idea.* First we prove the accuracy of Alg.3. It uses Alg. 1 with $\varepsilon = 1$ as a subroutine, thus we can get a set $\mathcal{K}$ such that $\mathcal{S}_L^{\rightarrow} \subseteq \mathcal{K} \subseteq \mathcal{S}_{2L}^{\rightarrow}$. For each goal state $g$, when $g$ is removed from $\mathcal{U}$ and the policy $\pi_g$ is set, the cost to reach $g$ from $s_0$ under policy $\pi_g$ has been tested for $\lambda = \widetilde{O}(1/\epsilon^2)$ times. Therefore, by the concentration inequality, we can prove that the average cost is close to the expected cost of $\pi_g$. Moreover, the average cost is less than our estimation of the cost of the optimal policy $V_g(s_0)$ plus $\epsilon L$. Hence we can prove that the expected cost of $\pi_g$ is close to the optimal cost using the optimism property.

To bound the cumulative cost of Alg.3, we will focus on the number of "rounds" and the cost it takes in each round. By multiplying them together, we can get the upper bound of the cumulative cost.

It's straightforward to show that the total cost in each round is no larger than $\widetilde{O}(L/\varepsilon^2)$. Then we will bound the total number of rounds. We denote $T$ as the total number of samples collected in Alg. 3. The number of success rounds is at most $K$. And the trigger condition holds for at most $\log_2(2T)$ times for each state-action pair $(s, a)$, thus we obtain that the number of skipped rounds can be bounded by $K'A \log_2(2T)$. Now we need only to bound the total number of failure rounds. We borrow the idea from (Lim and Auer, 2012). We first define the regret of an episode as the total cost in this episode minus our estimation of the optimal cost, and define the total regret as the sum of the regret in each episode. Then we prove a regret bound of our Alg. 3, and the upper bound grows sublinearly with the total number of episodes $M$. As the regret in each failure round is at least $\epsilon L \lambda$, thus we can give a lower bound of the total regret that grows linearly with the number of episodes $M$. By solving the inequality, we will obtain that the total number of failure rounds is bounded by $\widetilde{O}(KA + \varepsilon K^2 A)$. Thus the total number of rounds is bounded by $\widetilde{O}(KA + \varepsilon K^2 A)$, and multiplying it with the cost in each round, we can get the bound in Thm. 3 and complete the proof.

$\square$

Now we give the full proof. Here we denote $V_g^*(s)$ as the expected cost of the optimal policy to reach $g$ from $s$ on MDP $M^\dagger$, and we have $V_g^*(s) \le 2L + 1 \le B$ for all $(s, g) \in \mathcal{K}^\dagger \times \mathcal{K}$.

We recall the definition of the high-probability event $\mathcal{E}$ in Appendix E. Also, we introduce the lemma of optimism and the lemma of bounded error in Alg. 3, and the proof of them is the same with Lem. 12 and Lem. 13, respectively.

**Lemma 14.** *In Alg.3, under the event $\mathcal{E}$, for any output $(Q, V)$ of the* `VISGO` *procedure, it holds that*

$$Q(s, a) \le Q_g^*(s, a), \qquad\qquad \forall (s, a) \in \mathcal{K}^\dagger \times \mathcal{A},$$
$$V(s) \le V_g^*(s), \qquad\qquad \forall s \in \mathcal{K}^\dagger,$$

*where $g$ is the goal state in that round.*

**Lemma 15.** *In Alg. 3, under the event $\mathcal{E}$, for any output $(Q, V)$ of the* `VISGO` *procedure, let $\pi$ be the greedy policy with respect to $Q$. Then $\pi$ is proper on the model $P_{s,a,s'}$, and for all $s \in \mathcal{K}^\dagger$, we have $V_g^\pi(s) \le 2V(s)$, where $g$ is the goal state in that round.*

Now we prove the correctness of Alg. 3, i.e. Alg. 3 satisfies $AX^*$ on $\mathcal{K}$. The main intuition is that each policy $\pi_s$ has been tested for $\lambda$ times in a success round, and the average cost it takes to reach $s$ from $s_0$ is less than our estimate for optimal cost $V(s_0)$ plus $\epsilon L$. Thus by concentration inequalities, the expected cost of $\pi_s$ is close to optimal.

**Lemma 16.** *Let $\{\pi_s\}_{s \in \mathcal{K}}$ be the set of policies output by Alg.3. With probability at least $1 - \delta$, $V_s^{\pi_s}(s_0) \le V_s^*(s_0) + \varepsilon L$ for all $s \in \mathcal{K}$.*

*Proof.* We fix any state $s \in \mathcal{S}$. In any given round where the chosen target is $s$, let $\hat{\tau}_k$ be the total cost in the $k$-th episode of that round. Recall that for the algorithm to output a policy $\pi_s$, its empirical performance after $\lambda$ episodes must satisfy that $\hat{\tau} \le V(s_0) + \epsilon L$, where $\hat{\tau} = \frac{\sum_{k=1}^\lambda \hat{\tau}_k}{\lambda}$.

We define the random variable $\tau$ as the total cost it takes to reach the goal state $s$ from the start state $s_0$ when executing policy $\pi_s$, and we have $E(\tau) = V_s^{\pi_s}(s_0)$ by definition. We note that we have collected $\phi = \widetilde{O}(L^2/c_{\min}^2)$ samples for each of the state-action pair $(s, a)$. By Lem. 15, under event $\mathcal{E}$, the policy $\pi_s$ is proper, and we have $E(\tau) \le 2L$. Hence, we obtain $d := \|V_s^{\pi_s}\|_\infty \le 2L + 1 \le 4L$. By Lem 3, we obtain $\Pr(\tau > m) \le 2 \exp(-m/4d)$ for any $m > 0$.

As we set

$$\lambda = \lceil \frac{2048}{\epsilon^2} \ln^2(\frac{256}{\epsilon}) \ln(\frac{2S}{\delta}) \rceil,$$

by Lem. 4, we obtain that with probability at least $1 - \delta/(2S)$, we have

$$V_s^{\pi_s}(s_0) = E(\tau) \leq \hat{\tau} + \epsilon L.$$

We note that $\hat{\tau} \leq V(s_0) + \epsilon L \leq V_s^*(s_0) + \epsilon L$ by the optimism property (Lem. 14). Hence, as we set $\epsilon = \varepsilon/3$ in initial, we obtain that with probability at least $1 - \delta/(2S)$,

$$V_s^{\pi_s}(s_0) \leq V_s^*(s_0) + \varepsilon L.$$

Finally, as there are at most $S$ states in total, and the event $\mathcal{E}$ holds with probability at least $1 - \delta/2$, by the union bound, the total success probability is at least $1 - \delta$. $\qquad\square$

Now we focus on bounding the cumulative cost of Alg.3. The key idea lies in bounding the "regret". We will use the regret to bound the total number of rounds. We first define the regret in the $k$-th episode of the $j$-th round. We denote $H^{j,k}$ as the number of steps it takes in the $k$-th episode of the $j$-th round. The regret in an episode $k$ is defined as

$$(\sum_{h=1}^{H^{j,k}} c_h^{j,k}) - V^j(s_0),$$

where $c_h^{j,k}$ is the empirical cost in the $h$-th step in the $k$-th episode in the $j$-th round, and $V^j(s_0)$ is the value of $V(s_0)$ in the $j$-th round. Let $n_j$ be the total number of episodes executed in the $j$-th round, we define the regret in the $j$-th round as follows:

$$\sum_{k=1}^{n_j}((\sum_{h=1}^{H^{j,k}} c_h^{j,k}) - V^j(s_0)).$$

Then we will define the total regret of Alg.3. Let $r$ be the total number of rounds, $n_j$ be the total number of episodes executed in the $j$-th round, and $0 \leq n_j \leq \lambda$. Then we know that the total number of episodes in the whole process of Alg.3 is $M = \sum_{j=1}^{r} n_j$. For notation convenience, we define $H^m$ as the number of steps it takes in the $m$-th episode of the whole process of Alg.3, and denote $c_h^m$ as the empirical cost in the $h$-step of episode $m$. Finally we define the total regret of all the rounds as

$$R := \sum_{j=1}^{r}\sum_{k=1}^{n_j}((\sum_{h=1}^{H^{j,k}} c_h^{j,k}) - V^j(s_0)) = \sum_{m=1}^{M}((\sum_{h=1}^{H^m} c_h^m) - V^m(s_0)).$$

We will give both the upper bound and the lower bound of the regret. Here we gives the upper bound.

**Lemma 17.** *Under event $\mathcal{E}$, the total regret in $M$ episodes is at most*

$$R = \widetilde{O}(L\sqrt{KAM} + LK^2A).$$

This upper bound comes from the regret bound of the EB-SSP algorithm (Tarbouriech et al., 2021), which solves the classical SSP problem, and the proof of Lem. 17 is similar with Thm.3 in (Tarbouriech et al., 2021).

We observe that there are at most $\widetilde{O}(KA)$ skipped rounds and $K$ success rounds. We denote by $r_f$ the number of failure rounds, and we have the total number of episodes $M = \widetilde{O}((KA + r_f)\lambda) = \widetilde{O}((KA + r_f)/\epsilon^2)$. Thus the total regret in $r$ rounds can be bounded by $r_f$ sublinearly:

$$R = \widetilde{O}(\frac{L}{\epsilon}\sqrt{KAr_f} + \frac{LKA}{\epsilon} + LK^2A).$$

Then we gives the lower bound of the total regret in terms of the number of failure rounds $r_f$.

**Lemma 18.** *With probability $1 - \delta$, when $r = \widetilde{O}((SA)^2)$, the total regret in the $r$ rounds is at least*

$$R = \widetilde{\Omega}(\frac{Lr_f}{\epsilon} - \frac{LKA}{\epsilon}),$$

*where $r_f$ in the number of failure rounds in the $r$ rounds.*

*Proof.* By the criterion of our performance check, in any failure round, we have $\hat{\tau} > V(s_0) + \epsilon L$, and in round $j$, we have $\hat{\tau} = \frac{1}{\lambda} \sum_{k=1}^{n_j} (\sum_{h=1}^{H^{j,k}} c_h^{j,k})$ by definition. Hence, in any failure round $j$, the regret is $\lambda\hat{\tau} - n_j V^j(s_0) \geq \lambda(\hat{\tau} - V^j(s_0)) \geq \lambda\epsilon L = \widetilde{\Omega}(\frac{Lr_f}{\epsilon})$.

Then we focus on skipped rounds and success rounds. We denote $g^j$ as the goal state in the $j$-th round, and $\pi_j$ as the policy $\tilde{\pi}$ in the $j$-th round, which is the greedy policy over the $Q$-function in the $j$-th round. We observe that the regret in any round $j$ satisfies

$$\sum_{k=1}^{n_j}((\sum_{h=1}^{H^{j,k}} c_h^{j,k}) - V^j(s_0)) \geq -L + \sum_{k=1}^{n_j-1}((\sum_{h=1}^{H^{j,k}} c_h^{j,k}) - V_{g^j}^*(s_0)) \geq -L + \sum_{k=1}^{n_j-1}((\sum_{h=1}^{H^{j,k}} c_h^{j,k}) - V_{g^j}^{\pi_j}(s_0)),$$

where we used the optimism property in Lem. 14. We note that $\sum_{h=1}^{H^{j,k}} c_h^{j,k}$ is the empirical cost of policy $\pi_j$ in episode $k$, and we will use the concentration inequality to give a lower bound of the regret in round $j$. As the last episode in a skipped round can terminate before reaching the goal, we should take special considerations the last episode of each round. We directly use $-L$ to lower bound the regret of the last episode in round $j$. Then we denote $n = n_j - 1$, and focus on the previous $n$ episodes in round $j$.

Now we fix the round index $j$. We denote the random variable $\tau$ as the cost to reach $g^j$ from $s_0$, and we recall that $\hat{\tau}_k = \sum_{h=1}^{H^{j,k}} c_h^{j,k}$. By Lem. 4 with $d = 4L$, with probability at least $1 - \frac{\delta}{(SA)^2}$, we have

$$\sum_{k=1}^{n}(\hat{\tau}_k - E(\tau)) \geq -2\Gamma\sqrt{n \ln(\frac{SA}{\delta})} \geq -2\Gamma\sqrt{\lambda \ln(\frac{SA}{\delta})} \geq -\widetilde{\Omega}(\frac{L}{\epsilon}),$$

where $\Gamma = \lfloor 8d \ln(64/\epsilon) \rfloor$. Thus the regret in any round $j$ is no less than $-\widetilde{\Omega}(\frac{L}{\epsilon})$. As there are at most $\widetilde{O}(KA)$ skipped rounds and $K$ success rounds, we obtain that the total regret $R$ has the lower bound

$$R = \widetilde{\Omega}(\frac{Lr_f}{\epsilon} - \frac{LKA}{\epsilon}).$$

Now we bound the total failure probability. The number of rounds $r = \widetilde{O}((SA)^2)$, in each round the failure probability is at most $\frac{\delta}{(SA)^2}$, and the events $\mathcal{E}$ fails with probability $\delta/2$. By resetting $\delta \to O(\delta/(\frac{SAL}{\varepsilon}))$ in initial, the total failure probability is at most $\delta$. $\qquad\square$

As the lower bound is linear in $r_f$, and the upper bound is sublinear in $r_f$, we can solve it and obtain that $r_f = \widetilde{O}(KA + \epsilon K^2 A)$, thus the total number of rounds can be bounded by $\widetilde{O}(KA + \epsilon K^2 A)$.

To get the cumulative cost bound in Thm. 3, we need only to bound the cost in a round. In any round, we observe that except for the last episode, the average cost $\hat{\tau}$ for all the other episodes is no larger than $V(s_0) + \epsilon L \leq 2L$, thus the total cost in these episodes is no larger than $2L\lambda = \widetilde{O}(L/\epsilon^2)$. Also, we know that in the any episode, the expected cost of the policy $\tilde{\pi}$ to reach the goal from $s_0$ is no larger than $2L$. Thus by Lem. 3, in any round, with probability at least $1 - \frac{\delta}{(SA)^2}$, the cost in the last episode is no larger than $\widetilde{O}(L)$. Hence, the total cost in each round is no larger than $\widetilde{O}(L/\epsilon^2)$. By multiplying it with $\widetilde{O}(KA + \epsilon K^2 A)$, the cumulative cost in Alg.3 can be bounded by $\widetilde{O}(LKA/\varepsilon^2 + LK^2A/\varepsilon + L^3KA/c_{\min}^2)$, where the term $L^3KA/c_{\min}^2$ comes from the subroutine of Alg. 1. As we have $K \leq S_{2L}$, we obtain the bound in Thm. 3.

Now we count the total failure probability. First, Alg. 1 fails with probability $\delta/2$, the event $\mathcal{E}$ fails with probability $\delta/2$, and the lower bound of the total regret $R$ fails with probability $\delta$. And in the

previous para, to bound the cost in the last episode of each round using Lem. 3, the failure probability in each episode is at most $\frac{\delta}{(SA)^2}$. We observe that the total number of these failures is no larger than the total number of rounds, and the total number of rounds can be bounded by $\widetilde{O}(KA + \epsilon K^2 A)$, where we omit the logarithmic factors. Thus by setting $\delta \to O(\delta/(\frac{SAL}{\varepsilon}))$ in initial, we can bound the total failure probability by the initial $\delta$, and the proof of Thm. 3 is completed.

## G    ANALYSIS OF THE LOWER BOUND

Here we discuss the lower bound of the autonomous exploration problem. We recall that we relax the problem and consider only all the MDPs $\mathcal{M}$ with $S$ states and $A$ actions, and $\mathcal{M}$ satisfies Asmp. 1. Our algorithm needs output a set $\mathcal{K} \supseteq \mathcal{S}_L^{\rightarrow}$ and a set of policies $\{\pi_s\}_{s \in \mathcal{K}}$, and when $s \in \mathcal{S}_L^{\rightarrow}$, the policy $\pi_s$ satisfies $V_s^{\pi_s}(s_0) \leq (1+\varepsilon)L$. Moreover, we note that here we allow the algorithm to output non-deterministic policies, i.e. the policy $\pi_s : \mathcal{S} \to \Delta(\mathcal{A})$ is a function from the set of states to a probability distribution over the actions.

We recall the some basic concepts about the definition of a learning algorithm, and we use the notations in (Domingues et al., 2021). Let $\mathcal{I}^t = (\mathcal{S} \times \mathcal{A})^{t-1} \times \mathcal{S}$ be the set of all possible histories up to $t$ steps, that is, be the set of tuples of the form $\left(s^1, a^1, s^2, a^2, \ldots, s^t\right) \in \mathcal{I}^t$. We denoted $\boldsymbol{\pi} \triangleq (\pi^t)_{t \geq 1}$ as a history-dependent policy, where $\pi^t : \mathcal{I}^t \to \Delta(\mathcal{A})$ describes the probability of taking action $a \in \mathcal{A}$ after observing some history $i^t \in \mathcal{I}^t$.

Given an MDP $\mathcal{M} = \langle \mathcal{S}, \mathcal{A}, p, c, s_0 \rangle$, a policy $\boldsymbol{\pi}$ interacting with the MDP $\mathcal{M}$ defines a stochastic process denoted by $(S^t, A^t)_{t \geq 1}$, where $(S^t, A^t)$ is the state-action pair at time $t$. The Ionescu-Tulcea theorem ensures the existence of the probability space $(\Omega, \mathcal{F}, \mathbb{P}_{\mathcal{M}})$ such that

$$\mathbb{P}_{\mathcal{M}}\left[S^1 = s\right] = \mathbb{I}[s = s_0], \mathbb{P}_{\mathcal{M}}\left[S^{t+1} = s \mid A^t, I^t\right] = p\left(s \mid S^t, A^t\right), \text{ and } \mathbb{P}_{\mathcal{M}}\left[A^t = a \mid I^t\right] = \pi^t\left(a \mid I^t\right),$$

where $\boldsymbol{\pi} = (\pi^t)_{t \geq 1}$ and for any $t$, $I^t \triangleq \left(S^1, A^1, S^2, A^2, \ldots S^t\right)$ is the random vector in $\mathcal{I}^t$ containing all state-action pairs observed up to step $t$. We denote the $\sigma$-algebra generated by $I^t$ as $\mathcal{F}^t$. And we denote by $\mathbb{P}_{\mathcal{M}}^{I^T}$ the measure of $I^T$ under $\mathbb{P}_{\mathcal{M}}$ as follows:

$$\mathbb{P}_{\mathcal{M}}^{I^T}\left[i^T\right] \triangleq \mathbb{P}_{\mathcal{M}}\left[I^T = i^T\right] = \mathbb{I}(s^1 = s_0) \prod_{t=1}^{T-1} \pi^t\left(a^t \mid i^t\right) p\left(s^{t+1} \mid s^t, a^t\right).$$

Then we denote $\mathbb{E}_{\mathcal{M}}$ as the expectation under $\mathbb{P}_{\mathcal{M}}$. Note that the dependence of $\mathbb{P}_{\mathcal{M}}$ and $\mathbb{E}_{\mathcal{M}}$ on the policy $\boldsymbol{\pi}$ is denoted implicitly in the definition of $\mathbb{P}_{\mathcal{M}}$. We will denote them explicitly as $\mathbb{P}_{\boldsymbol{\pi}, \mathcal{M}}$ and $\mathbb{E}_{\boldsymbol{\pi}, \mathcal{M}}$ respectively when we need to stress $\boldsymbol{\pi}$.

Moreover, given the MDP $\mathcal{M} = \langle \mathcal{S}, \mathcal{A}, p, c, s_0 \rangle$, we denote $V_{\mathcal{M},g}^*(s)$ as the optimal policy to reach the goal state $g$ from the state $s$.

Here we introduce the basic definitions and the technical lemmas used in our proof.

**Definition 7.** *The Kullback-Leibler divergence between two distributions $\mathbb{P}_1$ and $\mathbb{P}_2$ on a measurable space $(\Omega, \mathcal{G})$ is defined as*

$$\mathrm{KL}(\mathbb{P}_1, \mathbb{P}_2) \triangleq \int_{\Omega} \log\left(\frac{d\mathbb{P}_1}{d\mathbb{P}_2}(\omega)\right) d\mathbb{P}_1(\omega),$$

*if $\mathbb{P}_1 \ll \mathbb{P}_2$ and $+\infty$ otherwise. For Bernoulli distributions, we define $\forall (p, q) \in [0,1]^2$,*

$$\mathrm{kl}(p, q) \triangleq \mathrm{KL}(\mathcal{B}(p), \mathcal{B}(q)) = p \log\left(\frac{p}{q}\right) + (1-p) \log\left(\frac{1-p}{1-q}\right).$$

**Lemma 19** (Lemma 5, Domingues et al. (2021), modified)**.** *Let $\mathcal{M}$ and $\mathcal{M}'$ be two MDPs that are identical except for their transition probabilities, denoted by $p$ and $p'$, respectively. Assume that we have $\forall (s, a)$, $p(\cdot \mid s, a) \ll p'(\cdot \mid s, a)$. Then, for any stopping time $\tau$ with respect to $(\mathcal{F}^t)_{t \geq 1}$ that satisfies $\mathbb{P}_{\mathcal{M}}[\tau < \infty] = 1$,*

$$\mathrm{KL}\left(\mathbb{P}_{\mathcal{M}}^{I^\tau}, \mathbb{P}_{\mathcal{M}'}^{I^\tau}\right) = \sum_{s \in \mathcal{S}} \sum_{a \in \mathcal{A}} \mathbb{E}_{\mathcal{M}}\left[N_{s,a}^\tau\right] \mathrm{KL}(p(\cdot \mid s, a), p'(\cdot \mid s, a)),$$

where $N_{s,a}^{\tau} \triangleq \sum_{t=1}^{\tau} \mathbb{1}\{(S^t, A^t) = (s,a)\}$ and $I^{\tau}$ is the random vector representing the history of $\tau$ samples.

**Lemma 20** (Lemma 1, Garivier et al. (2019))**.** *Consider a measurable space $(\Omega, \mathcal{F})$ equipped with two distributions $\mathbb{P}_1$ and $\mathbb{P}_2$. For any $\mathcal{F}$-measurable function $Z : \Omega \to [0,1]$, we have*

$$\mathrm{KL}(\mathbb{P}_1, \mathbb{P}_2) \geq \mathrm{kl}(\mathbb{E}_1[Z], \mathbb{E}_2[Z]),$$

*where $\mathbb{E}_1$ and $\mathbb{E}_2$ are the expectations under $\mathbb{P}_1$ and $\mathbb{P}_2$ respectively.*

**Lemma 21.** *For any $p, q \in (0, \frac{1}{2}]$, $\mathrm{kl}(p, q) \leq \frac{2(p-q)^2}{q}$.*

Now we construct a family of adversarial MDPs to obtain the lower bound of sample complexity.

**An example of hard MDP with two states.** For clarity, we first consider an MDP with two states $s$ and $g$. There are $A$ actions in the state $s$, and each action $a$ transits $s$ to $g$ with success probability $((1+\varepsilon)L)^{-1}$, and transits $s$ to $s$ with failure probability $1 - ((1+\varepsilon)L)^{-1}$. In this MDP, the expected cost to reach $g$ from $s$ is $(1+\varepsilon)L$. Then we construct $A$ MDPs, each MDP increasing the success probability of an action $a$ to $L^{-1}$, and the the expected cost to reach $g$ from $s$ is $L$ in all of these MDPs. Thus to discriminate these MDPs, we need to discriminate between the two Bernoulli distributions with $p = L^{-1}$ and $p = ((1+\varepsilon)L)^{-1}$. Using the KL-divergence, we can obtain that the minimum number of samples is $\widetilde{O}(L/\varepsilon^2)$. As there are $A$ actions, we can obtain the $\widetilde{O}(LA/\varepsilon^2)$ lower bound.

**The construction of hard MDPs with general $S_L$.** Now we fix $L, A, S_L, \varepsilon$ such that $L > 4$, $A > 4$, $4 \leq S_L \leq \min\{(\frac{A}{2})^{\lfloor \frac{L}{2} \rfloor}, \frac{S}{2}\}$, $0 < \varepsilon < \frac{1}{4}$. It's straightforward to verify that we can construct an MDP $\mathcal{M}_0' = \langle \mathcal{S}', \mathcal{A}', p_0', c, s_0 \rangle$ with $|\mathcal{S}'| = S_L - 1$ states and $|\mathcal{A}'| = \lceil A/2 \rceil$ actions, such that $c(s,a) = c_{\min}$ for each $(s,a) \in \mathcal{S}' \times \mathcal{A}'$, and each state $s \in \mathcal{S}'$ is incrementally $(L/2)$-controllable, i.e. its $\mathcal{S}_{L/2}^{\to} = \mathcal{S}'$. Thus for each $s \in \mathcal{S}'$, we have $V_{\mathcal{M}_0', s}^*(s_0) \leq \frac{L}{2}$.

Then we will construct the MDP $\mathcal{M}_0 = \langle \mathcal{S}, \mathcal{A}, p_0, c, s_0 \rangle$ based on $\mathcal{M}_0'$ in the following way. First we construct $S_L - 1$ states and $\lfloor A/2 \rfloor$ actions, and denote them as $\mathcal{S}^{\dagger}$ and $\mathcal{A}^{\dagger}$, respectively, and we set $\mathcal{S} = \mathcal{S}' \cup \mathcal{S}^{\dagger}$, $\mathcal{A} = \mathcal{A}' \cup \mathcal{A}^{\dagger}$. For each $s \in \mathcal{S}^{\dagger}$ and any actions $a \in \mathcal{A}$, we have $p_0(s_0|s,a) = 1$, i.e. every action at $s \in \mathcal{S}^{\dagger}$ is the reset action to $s_0$. And we set $c(s,a) = c_{\min}$ for each $(s,a) \in \mathcal{S} \times \mathcal{A}$. Moreover, for all $(s,a) \in \mathcal{S}' \times \mathcal{A}'$, our transition model $p_0(\cdot|s,a)$ is constructed based on $p_0'(\cdot|s,a)$, i.e. we set $p_0$ as follows:

$$
\begin{aligned}
p_0(s'|s,a) &= p_0'(s'|s,a), &\quad \forall (s,a,s') \in \mathcal{S}' \times \mathcal{A}' \times \mathcal{S}', \\
p_0(s'|s,a) &= 0, &\quad \forall (s,a,s') \in \mathcal{S}' \times \mathcal{A}' \times \mathcal{S}^{\dagger}, \\
p_0(s'|s,a) &= 0, &\quad \forall (s,a,s') \in \mathcal{S}' \times \mathcal{A}^{\dagger} \times \mathcal{S}',
\end{aligned}
$$

and then we need only define the transition function $p_0(g|s,a)$ for each $(s,a,g) \in \mathcal{S}' \times \mathcal{A}^{\dagger} \times \mathcal{S}^{\dagger}$. We note that as $|\mathcal{S}^{\dagger}| = S_L - 1$, for each $g \in \mathcal{S}^{\dagger}$, we can pair it with a state $s \in \mathcal{S}'$, and all pairs $(g,s)$ are mutually disjoint. Thus we use the notation $g_s \in \mathcal{S}^{\dagger}$ to denote a state in $\mathcal{S}^{\dagger}$ that is paired with the state $s \in \mathcal{S}'$. Then for all $(s,a) \in \mathcal{S}' \times \mathcal{A}^{\dagger}$, we define

$$p_0(g_s|s,a) = \frac{c_{\min}}{(1+3\varepsilon)L - V_{\mathcal{M}_0', s}^*(s_0)}, p_0(s|s,a) = 1 - p_0(g_s|s,a),$$

i.e. each new action $a$ either transits $s \in \mathcal{S}'$ to its pair $g_s \in \mathcal{S}^{\dagger}$ or remains at $s$ itself.

We observe that $V_{\mathcal{M}_0, s}^*(s_0) = V_{\mathcal{M}_0', s}^*(s_0)$ for all $s \in \mathcal{S}'$, and $V_{\mathcal{M}_0, s}^*(s_0) = (1+3\varepsilon)L$ for all $s \in \mathcal{S}^{\dagger}$.

Then we define the adversarial examples of MDPs. Now we fix any state-action pair $(s,a) \in \mathcal{S}' \times \mathcal{A}^{\dagger}$. The adversarial MDP $\mathcal{M}_{(s,a)} = \langle \mathcal{S}, \mathcal{A}, p_{s,a}, c, s_0 \rangle$ is constructed from $\mathcal{M}_0$, and the only difference of $\mathcal{M}_{(s,a)}$ and $\mathcal{M}_0$ lies on the transition probability at the fixed state-action pair $(s,a)$, and we define $p_{s,a}(\cdot|s,a)$ as follows:

$$p_{s,a}(g_s|s,a) = \frac{c_{\min}}{L - V_{\mathcal{M}_0', s}^*(s_0)}, p_{s,a}(s|s,a) = 1 - p_{s,a}(g_s|s,a),$$

i.e. at state $s$, we slightly increase the probability of the action $a$ to reach the pair state $g_s$. And for all the state-action pairs $(s', a')$ other than $(s,a)$, the transition probability $p_{s,a}(\cdot|s', a')$ is the

same as $p_0(\cdot|s', a')$. Therefore, we have $V^*_{\mathcal{M}_{(s,a)}, g_s}(s_0) = L$, and $V^*_{\mathcal{M}_{(s,a)}, s'}(s_0) = V^*_{\mathcal{M}_0, s'}(s_0)$ for all $s' \in \mathcal{S} \setminus \{g_s\}$.

Finally, we define the family of our adversarial MDPs as $\{\mathcal{M}_0\} \cup \{\mathcal{M}_{(s,a)}\}_{(s,a) \in \mathcal{S}' \times \mathcal{A}^\dagger}$. We note that for each MDP $\mathcal{M}_{(s,a)}$, its $|\mathcal{S}^{\rightarrow}_L| = S_L$, and it satisfies Asmp. 1. Also, for the MDP $\mathcal{M}_0$, its $|\mathcal{S}^{\rightarrow}_L| = S_L - 1$, and it also satisfies Asmp. 1. Thus the family is valid for the relaxed autonomous exploration problem.

Now we give our proof of Thm. 5 through the adversarial family of MDPs. Here we use the techniques of Thm. 7 in (Tarbouriech et al., 2020).

*Proof.* We denote by $\mathbb{P}_{(s^*, a^*)} \triangleq \mathbb{P}_{\pi, \mathcal{M}_{(s^*, a^*)}}$ and $\mathbb{E}_{(s^*, a^*)} \triangleq \mathbb{E}_{\pi, \mathcal{M}_{(s^*, a^*)}}$ the probability measure and expectation in the MDP $\mathcal{M}_{(s^*, a^*)}$ by following $\pi$ and by $\mathbb{P}_0$ and $\mathbb{E}_0$ the corresponding operators in the MDP $\mathcal{M}_0$. We fix any algorithm $(\pi, \tau, \mathcal{K}, \{\pi_s\}_{s \in \mathcal{K}})$ that solves the AX problem. We will prove that when working on the MDP $\mathcal{M}_0$, the algorithm will cost at least $\Omega(\frac{LS_L A}{c_{\min} \varepsilon^2} \log \frac{1}{\delta})$ samples in expectation, i.e.

$$\mathbb{E}_0[\tau] = \Omega(\frac{LS_L A}{c_{\min} \varepsilon^2} \log \frac{1}{\delta}),$$

which yields that the lower bound of the total cost is $\Omega(\frac{LS_L A}{\varepsilon^2} \log \frac{1}{\delta})$.

Now we fix the state-action pair $(s^*, a^*) \in \mathcal{S}' \times \mathcal{A}^\dagger$, and we denote $g_{s^*} \in \mathcal{S}^\dagger$ as the state paired with $s^* \in \mathcal{S}'$. Also, we denote the random variable $N^\tau_{(s,a)}$ as the number of samples that the algorithm takes at the state-action pair $(s, a) \in \mathcal{S} \times \mathcal{A}$. For any $\mathcal{F}^\tau$-measurable random variable $Z$ taking values in $[0, 1]$, we have

$$\mathbb{E}_0\left[N^\tau_{(s^*, a^*)}\right] \frac{72 c_{\min} \varepsilon^2}{L}$$

$$\overset{(a)}{\geq} \mathbb{E}_0\left[N^\tau_{(s^*, a^*)}\right] \text{KL}\left(\frac{c_{\min}}{(1 + 3\varepsilon)L - V^*_{\mathcal{M}'_0, s^*}(s_0)}, \frac{c_{\min}}{L - V^*_{\mathcal{M}'_0, s^*}(s_0)}\right)$$

$$\overset{(b)}{=} \text{KL}\left(\mathbb{P}^{I^\tau}_0, \mathbb{P}^{I^\tau}_{(s^*, a^*)}\right)$$

$$\overset{(c)}{\geq} \text{KL}\left(\mathbb{E}_0[Z], \mathbb{E}_{(s^*, a^*)}[Z]\right),$$

where (a) uses Lemma 21 and $V^*_{\mathcal{M}'_0, s^*}(s_0) \leq \frac{L}{2}$; (b) uses Lemma 19; (c) uses Lemma 20.

For any policy $\pi : \mathcal{S} \rightarrow \Delta(\mathcal{A})$, we define $\pi(a|s)$ as the probability that the policy $\pi$ takes action $a$ at state $s$. Then for any $(s, a) \in \mathcal{S}' \times \mathcal{A}^\dagger$, we define the event $Z_{s,a} = \mathbb{1}\{$The algorithm's output satisfies $g_s \in \mathcal{K}$ and $\pi_{g_s}(a|s) \geq \frac{2}{3}\}$. And we set the event $Z = Z_{s^*, a^*}$. We observe that as the algorithm $(\pi, \tau, \mathcal{K}, \{\pi_s\}_{s \in \mathcal{K}})$ solves the AX problem, when working on the MDP $\mathcal{M}_{(s^*, a^*)}$, with probability at least $1 - \delta$, its output should satisfy $g_{s^*} \in \mathcal{K}$ and the expected cost of the policy $\pi_{g_{s^*}}$ satisfies $V^{\pi_{g_{s^*}}}_{s^*}(s_0) \leq (1 + \varepsilon)L$, which yields that $\pi_{g_{s^*}}(a^*|s^*) \geq \frac{2}{3}$. Therefore, for any $(s^*, a^*) \in \mathcal{S}' \times \mathcal{A}^\dagger$, we have

$$\mathbb{P}_{(s^*, a^*)}[Z_{s^*, a^*}] \geq 1 - \delta,$$

as the algorithm succeeds with probability at least $1 - \delta$.

Moreover, for any policy $\pi$ we observe that for any two state-action pairs $(s, a)$ and $(s, a')$, $a \neq a'$, we cannot have both $\pi(a|s) \geq \frac{2}{3}$ and $\pi(a'|s) \geq \frac{2}{3}$. Therefore, we obtain that given any state $s \in \mathcal{S}'$, we have

$$\sum_{a \in \mathcal{A}^\dagger} \mathbb{P}_0[Z_{s,a}] \leq 1,$$

as any two events $Z_{s,a}$ and $Z_{s,a'}$ $(a \neq a')$ are mutually exclusive.

We recall that we set $Z = Z_{s^*, a^*}$, and we can obtain

$$\text{kl}\left(\mathbb{E}_0[Z], \mathbb{E}_{(s^*, a^*)}[Z]\right) = \text{kl}\left(\mathbb{P}_0[Z_{s^*, a^*}], \mathbb{P}_{(s^*, a^*)}[Z_{s^*, a^*}]\right)$$

$$\overset{(a)}{\geq} (1 - \mathbb{P}_0[Z_{s^*,a^*}]) \log\left(\frac{1}{1 - \mathbb{P}_{(s^*,a^*)}[Z_{s^*,a^*}]}\right) - \log(2)$$

$$\overset{(b)}{\geq} (1 - \mathbb{P}_0[Z_{s^*,a^*}]) \log\left(\frac{1}{\delta}\right) - \log(2),$$

where (a) uses the fact that $\mathrm{kl}(p,q) \geq (1-p)\log\left(\frac{1}{1-q}\right) - \log(2)$ for any $p, q \in [0,1]$; (b) uses that $\mathbb{P}_{(s^*,a^*)}[Z_{s^*,a^*}] \geq 1 - \delta$. Therefore, we have

$$\mathbb{E}_0\left[N_{(s^*,a^*)}^{\tau}\right] \geq \frac{L}{72 c_{\min} \varepsilon^2}((1 - \mathbb{P}_0[Z_{s^*,a^*}]) \log\left(\frac{1}{\delta}\right) - \log(2)).$$

We recall that $\sum_{a \in \mathcal{A}^{\dagger}} \mathbb{P}_0[Z_{s,a}] \leq 1$. Thus we can obtain that for any $s^* \in \mathcal{S}'$,

$$\sum_{a^* \in \mathcal{A}^{\dagger}} \mathbb{E}_0\left[N_{(s^*,a^*)}^{\tau}\right] \geq \frac{L}{72 c_{\min} \varepsilon^2}((|\mathcal{A}^{\dagger}| - 1) \log\left(\frac{1}{\delta}\right) - \log(2)|\mathcal{A}^{\dagger}|).$$

Summing up all the state-action pairs $(s^*, a^*) \in \mathcal{S}' \times \mathcal{A}^{\dagger}$, we can obtain the lower bound of the sample complexity.

$$\mathbb{E}_0[\tau] = \sum_{(s,a) \in \mathcal{S} \times \mathcal{A}} N_{(s,a)}^{\tau} \geq \sum_{(s^*,a^*) \in \mathcal{S}' \times \mathcal{A}^{\dagger}} \mathbb{E}_0\left[N_{(s^*,a^*)}^{\tau}\right] \geq \frac{L|\mathcal{S}'|}{72 c_{\min} \varepsilon^2}((|\mathcal{A}^{\dagger}| - 1) \log\left(\frac{1}{\delta}\right) - \log(2)|\mathcal{A}^{\dagger}|)$$

As $|\mathcal{A}^{\dagger}| = \lfloor A/2 \rfloor$ and $|\mathcal{S}'| = S_L - 1$, provided that $L > 4$, $A > 4$, $4 \leq S_L \leq \min\{(\frac{A}{2})^{\lfloor \frac{L}{2} \rfloor}, \frac{S}{2}\}$, $0 < \varepsilon < \frac{1}{4}$, and $\delta < \frac{1}{16}$, we can eventually prove that

$$\mathbb{E}_0[\tau] = \Omega(\frac{L S_L A}{c_{\min} \varepsilon^2} \log \frac{1}{\delta}),$$

which yields the lower bound of the cumulative cost in Thm. 5. $\qquad\square$

## H    LOWER BOUNDS FOR MULTI-GOAL SSP

Here we formulate the lower bound $\mathcal{C}_{\mathrm{MSSP}}(L, S, \varepsilon, \delta)$ for the multi-goal SSP problem. First we define an algorithm for the multi-goal SSP problem as a triple $(\boldsymbol{\pi}, \tau, \{\pi_s\}_{s \in \mathcal{S}})$, which means the algorithm executes a history-dependent policy $\boldsymbol{\pi}$, and returns a set of policies $\{\pi_s\}_{s \in \mathcal{S}}$ after sampling $\tau$ times. Also, we allow $\pi_s$ to be non-deterministic policies.

**Definition 8.** *Given any $L \geq 1$, $0 < \varepsilon \leq 1$, $0 < \delta < 1$, and $S \in \mathbb{N}$, an algorithm $(\boldsymbol{\pi}, \tau, \{\pi_s\}_{s \in \mathcal{S}})$ is $(\varepsilon, \delta)$-PAC for the multi-goal SSP problem within cumulative cost $C$, if for any MDP $\mathcal{M} = \langle \mathcal{S}, \mathcal{A}, P, c, s_0 \rangle$ with at most $S$ states and $A$ actions such that $\mathcal{M}$ satisfies Asmp. 1 and $\mathcal{S}_L^{\rightarrow} = \mathcal{S}$, the algorithm always terminates after using $\tau$ samples, and the cumulative cost is always not larger than $C$, and with probability at least $1 - \delta$, it returns a set of policies $\{\pi_s\}_{s \in \mathcal{S}}$, such that $\forall s \in \mathcal{S}, V_s^{\pi_s}(s_0) \leq V_s^*(s_0) + \varepsilon L$.*

*The cumulative cost $\mathcal{C}_{\mathrm{MSSP}}(L, S, \varepsilon, \delta)$ for the multi-goal SSP problem is the least $C \in \mathbb{R}$ satisfying that there exists an algorithm $(\boldsymbol{\pi}, \tau, \{\pi_s\}_{s \in \mathcal{S}})$ within cumulative cost $C$ which is $(\varepsilon, \delta)$-PAC for multi-goal SSP.*

We remark that our constructed adversarial examples for the autonomous exploration problem can also be applied to multi-goal SSP using the similar proof with Thm. 5, i.e. $\mathcal{C}_{\mathrm{MSSP}}(2L, 2S, \varepsilon/2, \delta) \geq \mathcal{C}_{\mathrm{AX}}(L, S, \varepsilon, \delta)$. Thus we obtain the following lower bound for multi-goal SSP, which implies that our Alg. 3 is also minimax for multi-goal SSP problem.

**Theorem 6.** *Assume that $L > 8$, $A > 4$, $8 \leq S \leq (\frac{A}{2})^{\lfloor \frac{L}{2} \rfloor}$, $0 < \varepsilon < \frac{1}{8}$, and $\delta < \frac{1}{16}$. Then we have*

$$\mathcal{C}_{\mathrm{MSSP}}(L, S, \varepsilon, \delta) = \Omega(\frac{LSA}{\varepsilon^2} \log \frac{1}{\delta}).$$

