# OpenReview forum: "Near-Optimal Algorithms for Autonomous Exploration and Multi-Goal Stochastic Shortest Path"
_ICLR.cc/2022/Conference — ICLR 2022 Submitted_

### Official Review · Reviewer_hPxX · 2021-10-23

**Correctness:** 3
**Technical Novelty And Significance:** 2
**Empirical Novelty And Significance:** 1
**Recommendation:** 3
**Confidence:** 3

**Main Review:**

The paper studies an interesting topic and makes a significant contribution by presenting the first lower bound and improved upper bounds.
However, I have many problems with its presentation and technical novelty:
1. The authors did not provide the appendix in the supplementary material. Instead it is attached to the main paper in the same pdf file.
2. The multi-goal SSP problem is not presented clearly. In fact, I am not sure if I understood it correctly. Is this problem simply the autonomous exploration problem where all of the states of the MDP are $L$-controllable? If so, then this is not really an interesting problem since the authors (and previous work) already solve a more general problem. Moreover, I don't  understand why the name multi-goal SSP suits this problem since it does not seem like a generalization of SSP. Isn't this problem simply reward-free RL?
3. The authors refer to SSP quite a lot, but they never introduced it properly. The introduction of the multi-goal SSP problem is not clear, but maybe it could be clearer if the authors presented the standard SSP problem first. Furthermore, the authors use many techniques from SSP (e.g., Tarbouriech et al. 2021) so they should really dedicate at least one paragraph to discussing the algorithmic ideas behind SSP algorithms and give the relevant references (all of these references can be found in Tarbouriech et al. 2021).
4. The technical novelty in the algorithms is not clear to me. Algorithms 1 and 2 look exactly like the algorithms of Tarbouriech et al. 2020, and the only difference is that they use a tighter optimistic planning algorithms (the VISGO algorithm of Tarbouriech et al. 2021). Are there any other new features or difficulties in the analysis? It makes sense that the upper bound is better than Tarbouriech et al. 2020 by a factor of $S$ (actually I think it should also save a factor of $L$) but it does not seem like there is anything new here. If there are technical difficulties then the authors must highlight them and present the ways that they deal with them. In addition, I think that algorithm 3 does not have any real benefits. It trades a factor of $L^2 S_{(1+\epsilon)L}$ for a factor of $S_{2L}$, but is it really better? To me it looks like (in most actual applications) $S_{2L}$ would be at least of order $(S_L)^2$ and definitely larger than $L$.
5. The lower bound sketch is not clear at all. In my opinion it should have a lot more focus since this is the first lower bound for the problem. After reading the proof in the appendix, the construction looks almost the same as the lower bound for SSP in "Near-optimal regret bounds for stochastic shortest path" (Rosenberg et al. 2020). Are there real differences here? Is the required analysis different? I am having a hard time understanding why this lower bound does not simply arise from the SSP lower bound.
6. The authors claim that Tarbouriech et al. 2020 study uniform cost of 1, but it looks like section 4.3 in their paper they extend the results to non-uniform cost. Can the authors please comment on that? I would also be interested in the authors opinions regarding costs that may be zero (i.e., $c_{min}=0$). In SSP, strictly positive costs are a much easier case (almost similar to uniform cost of 1), so I wonder how can zero costs be handled in autonomous exploration.

**Summary Of The Paper:**

This paper studies the incremental autonomous exploration problem and introduces algorithms with improved sample complexity.
It also provides the first lower bound for the problem.

**Summary Of The Review:**

The paper studies an interesting topic and give improved sample complexity guarantees. However, the authors fail to present their contributions, novelties, difficulties and techniques clearly.

---

> ### Author Response · Authors · 2021-11-22
> **Response to Reviewer hPxX (Part 2)**
>
> >4. The technical novelty in the algorithms is not clear to me. Algorithms 1 and 2 look exactly like the algorithms of Tarbouriech et al. 2020, and the only difference is that they use a tighter optimistic planning algorithms (the VISGO algorithm of Tarbouriech et al. 2021). Are there new features or difficulties in the analysis?
>
> The main novelty is as follows:
>
> 1. First, in Algorithm 1 we use concentration on $(P-\hat{P})V^*_{K,g}$ (value-optimistic) instead of $|P-\hat{P}|_1$ (model-optimistic). The aim is to use the variance information of $V^*$, so that we can obtain a tighter bound and remove an $S$ term.
>
> 2. The difficulty is that **K is dependent on previous samples**, hence $V^*_{K,g}$ depends on the set $\hat{P}$. Trivially applying union bound on $K$ will incur a $log(2^S) = O(S)$ term, because there are exponentially many possible $K$. Hence, we incrementally define a series of sets $K_0, K_1, …, K_Z$ in p11, Def 4, and we only use concentration on these $K_i$. And we also need to modify the optimism property, because we cannot ensure that our estimation $V \leq V_K^*$ , we can only ensure that $V \leq V_{K_i}^*$.
>
> >To me it looks like (in most actual applications) S_{2L} would be at least of order (S_L)^2 and definitely larger than S_L.
>
> **We disagree.**
>
> First, in our paper and previous papers (Lim and Auer (2012), Tarbouriech et al. 2021), **it is implicitly assumed that that $S_L$ grows polynomially with $L$**, i.e. there exists a constant $d$, such that $S_{L} \leq L^d$. This is because all papers ignore the logarithmic factors on $S_L$, and if $S_L$ grows super-polynomially, one needs to study the dependency on $\log (S_L)$ more carefully.
> When $S_L$ grows polynomially with $L$, we have $S_{2L} \leq (2L)^d = 2^d L^d$. Since $2^d$ is also a constant, $S_{2L}$ has the same order with $L^d$.
>
>
>
> >5. The lower bound sketch is not clear at all. In my opinion it should have a lot more focus since this is the first lower bound for the problem. After reading the proof in the appendix, the construction looks almost the same as the lower bound for SSP in "Near-optimal regret bounds for stochastic shortest path" (Rosenberg et al. 2020). Are there real differences here? Is the required analysis different? I am having a hard time understanding why this lower bound does not simply arise from the SSP lower bound.
>
> Our construction differs from the hard instance proposed by Rosenberg et al. (2020) in the following way: We equally divide the state set $S$ into two sets $S’$ and $S^\dagger$. For each state $s_i$ in $S’$, we pair it with a state $g_i$ in $S^\dagger$ and we add an “edge” from $s_i$ to $g_i$.
> The aim of the pairing is to ensure that the only way to reach $g_i$ is to first reach $s_i$ and then reach $g_i$ by some action $a$. With this pairing, we can ensure that the optimal policy with goal $g_i$ is determined by the optimal policy with goal $s_i$, which is essential for our proof.
>
> >6. The authors claim that Tarbouriech et al. 2020 study uniform cost of 1, but it looks like section 4.3 in their paper they extend the results to non-uniform cost. Can the authors please comment on that? I would also be interested in the authors opinions regarding costs that may be zero (i.e., c_min = 0). In SSP, strictly positive costs are a much easier case (almost similar to uniform cost of 1), so I wonder how can zero costs be handled in autonomous exploration.
>
> Although Tarbouriech et al. 2020 claims that their algorithm DisCo claims that their algorithm can be extended the results to non-uniform cost in Sec 4.3, their cost function $c(s,a)$ still has to be a “constant” in $[c_{min},1]$ in their analysis in Appendix B and C. In our analysis, we introduced concentrations on both $P$ and $c$. Hence, our analysis generalizes the autonomous exploration problem to the case that $c(s,a)$ is a random variable on $[c_{min},1]$.
>
> Zero cost is difficult to be handled in autonomous exploration, hence all previous algorithms mainly focus on the case that $c_{min} = 1$. This is because in autonomous exploration, we use “total number of samples” instead of “regret” to evaluate our algorithm. In the classical SSP, the total number of samples $T$ often has an  $1/c_{min}$ dependency. Special treatments (such as perturbation on the reward to make reward strictly positive) are needed to deal with $c_{min} = 0$, and often will make the regret bounds worse than $\sqrt{T}$.
>
> We hope our response clarifies your concerns!

---

> ### Author Response · Authors · 2021-11-22
> **Response to Reviewer hPxX**
>
> Thank you a lot for your instructive review! Please find our response to your comments below.
>
> >1. The authors did not provide the appendix in the supplementary material. Instead it is attached to the main paper in the same pdf file.
>
> Attaching the appendix in the main paper is allowed in ICLR 2022. See Q2 of FAQ: https://iclr.cc/Conferences/2022/AuthorGuide
>
> >2. The multi-goal SSP problem is not presented clearly. In fact, I am not sure if I understood it correctly. Is this problem simply the autonomous exploration problem where all of the states of the MDP are L-controllable? If so, then this is not really an interesting problem since the authors (and previous work) already solve a more general problem. Moreover, I don't understand why the name multi-goal SSP suits this problem since it does not seem like a generalization of SSP. Isn't this problem simply reward-free RL?
>
> Yes, multi-goal SSP problem can be viewed as the autonomous exploration problem where all of the states of the MDP are $L$-controllable. We introduce this problem because our algorithm 3 works in this way: Algo 3 first uses Algo 1 to compute a set $K$ such that $K$ contains $S_L$ and $K$ is $(1+\epsilon)L$-controllable, then Algo 3 fixes the set $K$ and only focus on the policies restricted on $K$, i.e. it solves multi-goal SSP problem on $K$. Hence, although multi-goal SSP is simpler than autonomous exploration, we still introduce this problem in Section 2.
>
>
> **Multi-goal SSP is fundamentally different from Reward-free RL**:
>
> To see this, there is an $\Omega(S^2A/\epsilon^2)$ lower bound for reward-free but multi-goal SSP’s upper bound scales $O(SA/\epsilon^2)$ (for sufficiently small $\epsilon$), so these two problems cannot be equivalent
>
> Second, the setting is different, because reward-free RL contains two phases: exploration phase and planning phase. In exploration phase we have no knowledge of reward $r$, and in planning phase we cannot interact with MDP. But in multi-goal SSP, we can estimate the cost function $c$, and the agent does not need to separate into two phases.
>
> Lastly, we believe multi-goal SSP suits this name, because classical SSP has only one goal $g$, and multi-goal SSP has multiple goal states.
>
>
> >3. The authors refer to SSP quite a lot, but they never introduced it properly. The introduction of the multi-goal SSP problem is not clear, but maybe it could be clearer if the authors presented the standard SSP problem first. Furthermore, the authors use many techniques from SSP (e.g., Tarbouriech et al. 2021) so they should really dedicate at least one paragraph to discussing the algorithmic ideas behind SSP algorithms and give the relevant references (all of these references can be found in Tarbouriech et al. 2021).
>
> Here we compare Algorithm 3 (VALAE) with EB-SSP algorithm. Line 16-25 in the VALAE algorithm is similar with EB-SSP. The novelty of the VALAE algorithm compared with EB-SSP is as follows:
>
> a. In each “round” in line 9, we fix the greedy policy $\tilde{\pi}$ and execute $\lambda$ episodes. In each round we compute the average cost of policy $\tilde{\pi}$, denoted as $\hat{\tau}$. The novelty here is that we carefully set $\lambda = \tilde{O}(1/\epsilon^2)$, so that when $\hat{\tau}$ is no more than $(1+\epsilon)L$, by concentration inequalities (Lemma 16), we can guarantee that the expected cost of policy $\tilde{\pi}$ is bounded by $(1+3\epsilon)L$.
>
> b. The second novelty is in line 6, where we prepare $\phi = \tilde{O}(L^2)$ samples for each state-action pair $(s,a) \in \mathcal{K}\times\mathcal{A}$. This is to guarantee that in each round, the expectation of the greedy policy $\tilde{\pi}$ is no more than $\tilde{O}(L)$, (i.e., the VISGO in line 11 outputs a near-optimal policy up to a constant). This property is essential for us to apply the concentration inequality (Lemma 16).
>
> Combined with the two novelties above, our algorithm VALAE has a near-optimal sample complexity $\tilde{O}(LSA/\epsilon^2)$.

---

### Official Review · Reviewer_CNUB · 2021-11-01

**Correctness:** 3
**Technical Novelty And Significance:** 3
**Empirical Novelty And Significance:** 3
**Recommendation:** 6
**Confidence:** 3

**Main Review:**

# Pros and Cons

## Pros

- Proposed algorithms achieved better sample complexities on different criteria, as summarized in Table 1.  As I understand correctly, the four criteria are based on the existing literature but more clearly discussed and summarized in the four columns. Further, a restriction on the cost $c$ is relaxed by $c_\mathrm{min}$; that is, the authors generalize the problem settings.
- The proposed algorithm is well described (it is as complete and interpretable as those in two existing works UcbExplore and DisCo).

## Cons

- Some knowledge about autonomous exploration is used before the definition or references (i.e., Sec 1 is a bit complex without Sec 2 and reference papers).
- Readers are hard to follow all the details of theoretical proofs (much longer than the main content).
- No experimental results (cf., [Tarbouriech et al. 2020] contains some experimental evaluations)

# Comments

(I) A part of the statement in this paper is unclear for me.

In [Tarbouriech et al. 2020], the authors seem to say that the sample complexity is $\tilde{O}\left(L^5 S_{L+\epsilon}\Gamma_{L+\epsilon}A/\epsilon^2\right)$ under the definition of $\mathrm{AX}^\star$ of $\forall s\in\mathcal{K}, v_{\pi\_s}(s\_0\to s) \leq V^\star\_{\mathcal{S}^{\to}\_L}(s\_0\to s) + \epsilon$ (I copied this equation from Def.5 of [Tarbouriech et al. 2020]).

This definition is a slight different from the submitted paper (i.e., $(1+\epsilon)L$ vs $L + \epsilon$ or $\epsilon$ vs $\epsilon L$ or ?). Then, I cannot follow the part of Table 1 (it says $\tilde{O}(L^3+S^2_{(1+\epsilon)L}A/\epsilon^2)$). I feel that such correspondence could be explained a bit more clearly because the complexity statements are core contributions of the paper for possible readers.

(II) Please clarify the assumption of algorithms.

In Alg.1, the input is an MDP $M=\langle\mathcal{S, A}, P, c, s_0\rangle$. However, from the _Learning Objective_ part, an agent has no prior knowledge of $P$ and $c$, then I wonder how we can input such an $M$ to Alg.1. I guess I misunderstand some parts of the algorithms.

[Just a comment] The mathematical notations (Sec 1.2) are used prior to the definition (Sec 2). This makes readers hard follow the 'overview' of the paper. One approach to mitigate this issue is the paper stats with Sec.2 (i.e., a similar way of Tarbouriech et al. 2020), but I can also understand the current form of the structure.

Some minor comments and questions.
- p2. $P_{s,a}$ and $\hat{P}_{s,a}$ are the true and estimated transitions. Are these evaluated for all $s, a$? or partially discovered states on $\mathcal{K}$?
- p3. $z\in \{0,1,\dots,Z-1\}$ to incrementally define $\mathcal{K}_z$?
- p3. notation $\max_{s\in\mathcal{S}} |X(S)| \to \max_{s\in\mathcal{S}} |X(s)|$ as |S| means the number of states
- p3. Please describe the semantics of $\mathbb{V}(X, Y)$. I wonder, is it $X(s)^2 Y(s)^2$? (why the square is only for $Y(s)$?).
- p18. Notation $A[n(s,a)]$ is not defined?

**Summary Of The Paper:**

This paper studies the sample complexity of learning algorithms for autonomous explorations (AX). Theoretical results showed that the proposed algorithm (VOISD, VOISD+Re-MG-SSP, VALAE) achieved better sample complexities than the existing two studies. The authors compared the methods by setting the uniform cost. Further, the proposed algorithm seems to have more general performance.

**Summary Of The Review:**

The statement of the problem is clearly stated, and good theoretical results are reported with substantial proof. As I read and check, algorithms seem to work. Then, I feel that the proposed result is valuable for the field.

A disadvantage of this paper is that most proofs are included in the appendix, and therefore it is hard to follow and validate the correctness of the paper. To be honest, I cannot validate the correctness of all proof. I feel that the paper should be published as a journal article (e.g., Theoretical Computer Science).

---

> ### Author Response · Authors · 2021-11-22
> **Response to Reviewer CNUB**
>
> We thank you for your constructive review. Please find our response to your comments below.
>
> > This definition is a slight different from the submitted paper. Then, I cannot follow the part of Table 1. I feel that such correspondence could be explained a bit more clearly because the complexity statements are core contributions of the paper for possible readers.
>
> Our definition of $\epsilon$ is the same with Lim and Auer (2012) and differs from Tarbouriech et al. (2020). For clarity, here we compare the differences of \epsilon on the definition $AX_L$ on $S_L$ and $AX^*$ on $S_L$. We denote $\epsilon$ as the definition of $\epsilon$ in Lim and Auer (2012), and $\epsilon’$ as the definition of $\epsilon$ in Tarbouriech et al. 2020.
>
> a. $AX_L$ and $AX^*$ under $\epsilon$ in Lim and Auer (2012) (multiplicative $\epsilon$):
>
> $AX_L$ on $S_L$: $\forall s \in S_L, V^{\pi_{s}}_s(s_0) \leq (1 + \epsilon)L$.
>
> $AX^*$ on $S_L$: $\forall s \in S_L$, $V^{\pi_{s}}_s(s_0)  \leq V^*(s_0) + \epsilon L $.
>
> b. $AX_L$ and $AX^*$ under $\epsilon’$ in Tarbouriech et al. 2020 (additive $\epsilon’$):
>
> $AX_L$ on $S_L$: $\forall s \in S_L, V^{\pi_{s}}_s(s_0) \leq L + \epsilon’$.
>
> $AX^*$ on $S_L$: $\forall s \in S_L, V^{\pi_{s}}_s(s_0) \leq V^*(s_0) + \epsilon’$.
>
> In short, the relation between $\epsilon$ and $\epsilon’$ is: $\epsilon’ = L * \epsilon$.
>
> Therefore, in Tarbouriech et al. 2020, the sample complexity of DisCo is $L^5 S_{L+\epsilon’} A / \epsilon’^2$. We need to set $\epsilon’ = L * \epsilon$, so that the sample complexity of DisCo is $L^3 S_{(1+\epsilon)L} A / \epsilon^2$, which is consistent with Table 1.
>
> We will clarify this in the final version.
>
> >  (II) Please clarify the assumption of algorithms. I wonder how we can input such an M to Alg.1.
>
> We do not actually input $M$ to our algorithm. “Input: $M$” means that our algorithm “works on MDP $M$”, and we do not need to input the model $P$ and cost function $c$.
>
> Assumption of our algorithm:
> Our agent does not know $P$ or $c$. The agent knows the constants $S$,$A$ and $c_{min}$. The MDP $M$ should contain an “RESET” action.
>
> The input $M$ for Alg.1 means the agent can interact with $M$. We will clarify this in the final version. Thanks for pointing out,
>
> Some minor comments and questions.
>
> > p2: P_{s,a} and \hat{P}_{s,a} are the true and estimated transitions. Are these evaluated for all s,a? or partially discovered states on K?
>
> $\hat{P}_{s,a}$ are only evaluated on discovered states on $K$. In our algorithm, we only consider all the policies restricted on $K$.
>
> > p3: z = 0,1,..,Z-1 to incrementally define K_z?
>
> Yes, the definition is in p11, Def 4. First we define $K_0 = {s_0}$. Then we define $K_{i+1}$ from $K_i$ in the following way: $K_{i+1} = K_i \cup $ \{ state $s \notin K_i$ : $s$ can be reached from $s_0$ with expected cost no more than $L$}.
> When $K_Z = K_{Z-1}$, we stop this process.
>
> > p3: notation max_{s \in S} |X(S)| \rightarrow max_{s \in S} |X(s)| as |S| means the number of states.
>
> Yes, thanks fir pointing out this typo.
>
> > p3. Please describe the semantics of V(X,Y). (why the square is only for Y(s)?).
>
> Here $X$ is a probability distribution on the state set $S$, i.e. $X(s) \geq 0$ and $\sum_{s\in S} X(s) = 1$.
>
> $Y$ is a vector on the state set $S$.
>
> We can view $Y$ as a random variable, and $V(X,Y)$ is the variance of $Y$ under probability distribution $X$. Hence, we know that the variance is $E(Y^2) - E(Y)^2 = \sum X(s)(Y(s)^2) – (\sum X(s)Y(s))^2$, which is consistent with our definition of $V(X,Y)$.
>
> > p.18 Notation A[n(s,a)] is not defined?
>
> It is $A * n(s,a)$.
>
>
> We hope our response clarifies your concerns!

---

> > ### Comment · Reviewer_CNUB · 2021-11-24
> > **Thank you for your updates**
> >
> > Dear authors,
> >
> > Thank you for your comments, which clarify what I misunderstood or I cannot follow during the reviewing process. I'd like to keep my score as the paper is well-written and the technical contributions are clarified by comparing them with existing literature (maybe in the revised version).

---

### Official Review · Reviewer_nNZN · 2021-11-02

**Correctness:** 3
**Technical Novelty And Significance:** 3
**Empirical Novelty And Significance:** Not applicable
**Recommendation:** 6
**Confidence:** 3

**Main Review:**

For the positive side, the expression and organization of the paper is good. It is comfortable to read the paper and understand the critical points made by authors. The series construction argument and the connection between AX and multi-goal SSP is also very interesting. The proposed lower bound complements the AX theory largely.

For the negative side, I have a few concerns:
1.	It appears to me that the VALAE algorithm is a simple extension of EB-SSP from single point SSP to multi-goal SSP on the MDP induced by the first two algorithms. This somehow reduces the novelty of the paper.
2.	The data collection procedure of the algorithms rely heavily on the RESET action, as if there are a generative model. I wonder if we can design some algorithms to direct the agent from any state back to $s_0$, rather than relying on a RESET action. As far as I’m concerned, such reliance drops a lot of data
3.	There are many examples and detailed explanations, reducing the readability of the paper. This makes readers miss the important conclusions, but stick to those descriptive sentences.
4.	I am not sure about the technical novelty of the hard instance constrcution. It appears that the construction is much similar to the one proposed by Rosenberg et al. (2020). You both follow the novel construction proposed in the paper of UCRL2. The constrcution also looks similar. Please make it clear if there is any new idea.

**Summary Of The Paper:**

This paper studies the autonomous exploration problem and multi-goal SSP problem, and proposes three algorithms with improved cumulative costs on 4 learning objectives. The authors also construct a hard instance and show the lower bound of the cumulative cost. Their bounds are optimal in terms of $L, A, \epsilon$. The main technical contributions are a new series construction of $\mathcal{K}$, the connection between autonomous exploration and multi-goal SSP, and the construction of the hard instance.



**Summary Of The Review:**

The paper is generally well written, though some paragraphs need shorten. The technical contributions are a little confusing to me, so I recommend the boardline reject.

---

> ### Author Response · Authors · 2021-11-22
> **Response to Reviewer nNZN**
>
>
> We thank you for  your instructive review! Please find our responses to your comments below.
>
> > 1. It appears to me that the VALAE algorithm is a simple extension of EB-SSP from single point SSP to multi-goal SSP on the MDP induced by the first two algorithms. This somehow reduces the novelty of the paper.
>
> Line 16-25 in the VALAE algorithm is similar with EB-SSP. The novelty of the VALAE algorithm compared with EB-SSP is as follows:
>
> a. In each “round” in line 9, we fix the greedy policy $\tilde{\pi}$ and execute $\lambda$ episodes. In each round we compute the average cost of policy $\tilde{\pi}$, denoted as $\hat{\tau}$. The novelty here is that we carefully set $\lambda = \tilde{O}(1/\epsilon^2)$, so that when $\hat{\tau}$ is no more than $(1+\epsilon)L$, by concentration inequalities (Lemma 16), we can guarantee that the expected cost of policy $\tilde{\pi}$ is bounded by $(1+3\epsilon)L$.
>
> b. The second novelty is in line 6, where we prepare $\phi = \tilde{O}(L^2)$ samples for each state-action pair $(s,a) \in \mathcal{K}\times\mathcal{A}$. This is to guarantee that in each round, the expectation of the greedy policy $\tilde{\pi}$ is no more than $\tilde{O}(L)$, (i.e., the VISGO in line 11 outputs a near-optimal policy up to a constant). This property is essential for us to apply the concentration inequality (Lemma 16).
>
> Combined with the two novelties above, our algorithm VALAE has a near-optimal sample complexity $\tilde{O}(LSA/\epsilon^2)$.
>
>
> > 2. The data collection procedure of the algorithms rely heavily on the RESET action, as if there are a generative model. I wonder if we can design some algorithms to direct the agent from any state back to s_0, rather than relying on a RESET action. As far as I’m concerned, such reliance drops a lot of data.
>
> The RESET action is introduced in the first algorithm of autonomous exploration in Lim and Auer (2012), and Tarbouriech et al. (2020) also uses the RESET action. Here I discuss why we need RESET action. In classical SSP with a single goal, we have an episodic learning protocol, and when each episode ends (i.e. the agent reaches the goal), the agent can “reset” to initial state $s_0$ and start a new episode.
>
> In autonomous exploration and multi-goal SSP problem, we cannot have episodic learning protocol like classical SSP, because we need to ensure that for each goal the agent learns a near-optimal policy. Therefore, each time when the agent arrives at any of the goal, the agent **has to** “reset” to s_0. (This is also the reason that we use “total number of samples” instead of “regret” to evaluate the learning algorithm.) Hence the RESET action is necessary in autonomous exploration problem.
>
> Lastly, we note the RESET is much weaker than the generative model assumption in the literature. With generative model, the agent can directly access to **any** state whereas RESET can only get back to the initial state.
>
>
> > 3. There are many examples and detailed explanations, reducing the readability of the paper. This makes readers miss the important conclusions, but stick to those descriptive sentences.
>
> Thanks for your suggestions. We will move some of the technical explanations to the appendix in the final version.
>
> > 4. I am not sure about the technical novelty of the hard instance constrcution. It appears that the construction is much similar to the one proposed by Rosenberg et al. (2020). You both follow the novel construction proposed in the paper of UCRL2. The constrcution also looks similar. Please make it clear if there is any new idea.
>
> Our construction differs from the hard instance proposed by Rosenberg et al. (2020) in the following way: We equally divide the state set S into two sets $S’$ and $S^\dagger$. For each state $s_i$ in $S’$, we pair it with a state $g_i$ in $S^\dagger$ and we add an “edge” from $s_i$ to $g_i$.
>
> The aim of the pairing is to ensure that the only way to reach $g_i$ is to first reach $s_i$ and then reach $g_i$ by some action a. With this pairing, we can ensure that the optimal policy with goal $g_i$ is determined by the optimal policy with goal $s_i$, which is essential for our proof.
>
>
>  We hope our response clarifies your concerns!

---

> > ### Comment · Reviewer_nNZN · 2021-11-29
> > **Thank you for your clarifications**
> >
> > Thanks for answering my questions and making clarifications.
> >
> > The novelty behind VALAE and the hard instance construction is convincing to me. About the RESET action, I understand the its necessity in autonomous exploration problem. Overall, I would like o raise my score to 6.

---

### Official Review · Reviewer_J6YN · 2021-11-02

**Correctness:** 2
**Technical Novelty And Significance:** 3
**Empirical Novelty And Significance:** Not applicable
**Recommendation:** 5
**Confidence:** 3

**Details Of Ethics Concerns:**

No concerns

**Main Review:**

The paper is globally well written (see specific comments). I suggest defining at the beginning the learning framework, the (\epsilon,\delta) conditions for the different criteria and the sample complexity. I am not convinced by the proofs (see specific comments): some proofs are missing and some points are not clear (see specific comments about the proof of Lemma 9 and Theorem 1). I did not check the proofs of the lower bound. I think that the characterization of the rate for the incremental autonomous exploration problem is a valuable contribution. On the other hand, I find the proposed algorithm a bit involved (see specific comment).

I could increase my score if my concerns about proofs are cleared up.

Main comments:
-The algorithm is somewhat involved and not very natural. I think that the authors could have put more effort in the description of the algorithm and in providing intuition.

-Can you provide preliminary experiments to illustrate how the proposed algorithms behave on very simple MDP?

-Can you provide the time complexity and space complexity of VALAE algorithm?

Specific comments:
-Introduction: the sentence "This procedure also resembles some biological learning processes" is a bit cryptic.

-P1, note 1: It seems that there is an extra L in the big O.

-P4, end of the page: the learning framework is not completely clear: how does the agent interact with the MDP? when does the agent stop and output the policies (is T chosen by the agent)?

-P5, end of the page: the sentence "Then for each action [..] \hat{c}(s,a)" is not clear at all.

-P6, Alg2, L5: typo "where x is defined".

-P8, top of the page: The first paragraph is hard to follow since we do not know at this point what are Lemma 15 and 16.

-P9, above Definition 3: maybe you could have defined an algorithm for AX/* before in the introduction.

-P11, Lemma 2: Which concentration inequalities? This lemma should be proved even if it is a classical proof.

-P12, Lemma 4: You need an extra assumption on the \tau_k such that independence to be able to apply Hoeffding inequality.

-P15, end of the proof of Lemma 9: Can you detail why | P_{s,a,s'}  - \tilde{P}_{s,a,s'} | \leq \frac{c_min \epsilon} {6(L+1)}, it is not
completely obvious for me. And to apply Lemma 3 you need a control in norm 1 and not in infinite norm so there is a S factor missing here?

-P17, proof of Theorem 1: can you detail why on \cG_2 you get \hat{P}_{s,a,g} \geq 3/4 L if \P_{s,a,g} \geq1/ L? Typo "previous paragraph".

-P17, end of the proof of Theorem 1: can you write explicitly what is the event that happens with probability 1-\delta/4 to collect the sample? Because a priori for a given state s the policy used to reach s is random. Furthermore to use that V^\pi \leq L you used Lemma 9 that is you conditioned by the event \cG. Thus it is not clear that conditioned don this event the samples remain independent/ have the same distribution in order to apply Lemm 3.  Thus you cannot do a simple union bound over the state as it seems to be done also over policies. Maybe you should rather concentrate a well-chosen martingale.

-P18, proof of Theorem 2: same remark as above.

-P19, Lemma 14,15: can you provide proof for Lemma 14 and 15? In particular, in Algorithm 3 the Visgo procedure is called with \epsilon_VI = 2^-j... and to prove these two lemma you need according to the proof of Lemma 9 and 10 a lower bound on the number of visits of all state-action pairs that scales with \epsilon_VI .

**Summary Of The Paper:**

The authors consider the incremental autonomous exploration problem. That is, the agent faces an MDP and wants to learn near-optimal goal-conditioned policies to reach the states that are L-controllable, i.e. incrementally reachable from an initial state s_0 within L steps in expectation. The learning procedure is the following: the agent collects transition by interacting with the MDP, decides to stop, and outputs a set of states (that should be the L-controllable states) and a policy for each state ( that should be the associated near-optimal goal-conditioned policy). The authors propose the Value-Aware Autonomous Exploration (VALAE) algorithms. It works in 3 phases: in the first one a sub-algorithm Value-Optimistic Incremental State Discovery (VOISD) aims at identifying the set of  L-controllable states then sub-algorithm Reduce Autonomous Exploration to Multi-Goal SSP (Re-MG-SSP) as burn-in sample collector, finally a procedure close to the one of UCBExplore by Lime and Auer (2012). They prove a bound on the sample complexity ( the number of steps before the algorithm stops) of order O(L S_{2L} A / \epsilon^2 ) where S_L is the number of  L-controllable states, A the number of actions, \epsilon the error tolerated for the goal-conditioned policy. This bound improves prior results by Lim and Auer (2012) and Tarbouriech et al. (2020) with respect to the dependence on \epsilon or L. The authors also proved a lower bound of order  \Omega(L S_{L} A / \epsilon^2 )  implying that  VALAE is nearly minimax optimal if S_L grows polynomially with L.

**Summary Of The Review:**

See the main review

---

> ### Author Response · Authors · 2021-11-22
> **Response to Reviewer J6YN (Part 2)**
>
> Specific Comments:
>
> >-P9, above Definition 3: maybe you could have defined an algorithm for AX/* before in the introduction.
>
> We note that to prove the lower bound, in definition 3 we only use the “$AX_L$ for $S_L$” criteria, which is looser than the definition of $AX^*$ in the introduction.
>
> >-P11, Lemma 2: Which concentration inequalities? This lemma should be proved even if it is a classical proof.
>
> Please refer to Definition 12 and Lemma 13 in “Stochastic Shortest Path: Minimax, Parameter-Free and Towards Horizon-Free Regret”, respectively. We will add more details. Thanks for pointing out.
>
>
> >-P12, Lemma 4: You need an extra assumption on the $\tau_k$ such that independence to be able to apply Hoeffding inequality.
>
> Yes, here we need the independence assumption. Thanks for pointing out,
>
> >-P15, end of the proof of Lemma 9: Can you detail why | P_{s,a,s'} - \tilde{P}_{s,a,s'} | \leq \frac{c_min \epsilon} {6(L+1)}, it is not completely obvious for me. And to apply Lemma 3 you need a control in norm 1 and not in infinite norm so there is a S factor missing here?
>
> As we set $\phi = \Omega(L^2/(c_{min}^2\epsilon^2))$ in line 4, Algo 1, under the event $G_2$, we have $|P{s,a,s’} - \hat{P}{s,a,s’}| \leq O(\sqrt{1/\phi}) = O(c_{min} \epsilon /L)$ because $n(s,a) \geq \phi$. Moreover, as we have $|\tilde{P}{s,a,s'} - \hat{P}{s,a,s’}| \leq 1/n(s,a) \leq 1/\phi$ in Algo 4 (VISGO), we obtain that $|P{s,a,s’} - \hat{P}{s,a,s’}| \leq O(c_{min} \epsilon /L)$. By choosing the correct constant in $\phi$, we obtain $|P{s,a,s'} - \tilde{P}{s,a,s'} | \leq \frac{c_{min} \epsilon} {6(L+1)}$.
>
> >-P17, proof of Theorem 1: can you detail why on \cG_2 you get \hat{P}{s,a,g} \geq 3/4 L if \P{s,a,g} \geq1/ L? Typo "previous paragraph".
>
> As we set $\phi = \Omega(L^2)$ in line 4, Algo 1, under event $G_2$, we have $|P{s,a,g} - \hat{P}{s,a,g}| \leq O(\sqrt{1/\phi}) = O(1/L)$ because $n(s,a) \geq \phi$. By choosing the correct constant in $\phi$, we obtain $|P{s,a,g} - \hat{P}{s,a,g}| \leq 1/(4L)$. Hence we have $\hat{P}{s,a,g} \geq 3/(4L)$ if $P{s,a,g} \geq 1/ L$.
>
> >-P17, end of the proof of Theorem 1: can you write explicitly what is the event that happens with probability 1-\delta/4 to collect the sample? Because a priori for a given state s the policy used to reach s is random. Furthermore to use that V^\pi \leq L you used Lemma 9 that is you conditioned by the event \cG. Thus it is not clear that conditioned don this event the samples remain independent/ have the same distribution in order to apply Lemm 3. Thus you cannot do a simple union bound over the state as it seems to be done also over policies. Maybe you should rather concentrate a well-chosen martingale.
>
> The event that happens with probability $1-\delta/4$ to collect the sample is the following sentence: “The total cost incurred by policy $\pi_s$ in line 11, Algo 1 is no more than $\tilde{O}(L * \phi * S) = \tilde{O}(L^3S/\epsilon^2)$.” Here we denote this event as $D$.
>
> Now we show the reason that we can use the union bound on s instead of $\pi_s$. We note that the aim of line 11 is to reach the state $s$ from $s_0$. And in Algo 1, we do not use the samples in line 11. Algo 1 only collects samples after reaching state s in line 12-13, and we only focus on the total cost incurred by $\pi_s$ in line 11.
>
> To obtain that the event $D$ happens with probability $1-\delta/4$ by Lemma 4, we only use this property: “In any round $r$, the expected cost of policy $\pi_s$ to reach state $s$ from $s_0$ in line 11, Algo 1 is close to the empirical average cost.” To obtain this property, we apply the Hoeffding’s inequality (Lemma 4) and we obtain “In any round $r$, if the expected cost of policy $\pi_s$ is no more than $O(L)$, the total cost incurred line 11 in round $r$ is no more than $\tilde{O}(L * \phi)$.” By union bound on the total number of rounds $S$, we obtain event $D$. And the success event (holds with probability $1-\delta$) is “The event $G$ and all the Hoeffding’s inequalities do not fail.”
>
> We will give more details in the final version.
>
> > P18, proof of Theorem 2:  same remark as above.
>
> Similarly with above.
>
> > P19, Lemma 14,15:
>
> Lem 14:
> The proof of optimism property (Lem. 10, Lem. 12) does not depend on the exact settings of $\epsilon_{VI}$. We can directly view Lem. 14 as a special case of Lem. 12 with $\epsilon = 1$.
>
> Lem 15:
> The proof of bounded error property (Lem. 9, Lem. 13) only uses that the initial $j_0$ is no less than $5+log(1/c_{min})$, which guarantees that all the $\epsilon_{VI}$ is no more than $2^{-j_0}$. Hence we can directly view Lem. 15 as a special case of Lem. 13 with $\epsilon = 1$.
>
> We hope our response clarifies your concerns!

---

> > ### Comment · Reviewer_J6YN · 2021-11-23
> > **Additional clarification**
> >
> > Thanks for answering my questions.
> >
> > There is still a point that is not completely clear. Could you provide an answer to the comment:
> >
> > -P15, end of the proof of Lemma 9:  To apply Lemma 3 (simulation lemma for SSP) you need a control in norm 1 and not in the infinite norm, is there an S factor missing here?

---

> ### Author Response · Authors · 2021-11-22
> **Response to Reviewer J6YN**
>
> Thank you for your detailed and constructive review! Please find our response to your comments below.
>
> Main comments:
> >-The algorithm is somewhat involved and not very natural. I think that the authors could have put more effort in the description of the algorithm and in providing intuition.
>
> We briefly summarize our algorithms as follows:
> Algorithm 1 (VOISD) works in this way: In each round, we consider all the policies restricted on the current $K$, and we hope to discover states not in $K$ that are $L$-controllable. To achieve this, we collect $\phi$ samples for all state-action pair in $K \times A$, such that our estimation of model $P$ is accurate enough. The algorithm terminates when we cannot extend the current $K$.
> Algorithm 2 is to meet the $AX^*$ criterion. Algo 2 recollect $\phi$ samples for all state-action pair in $K \times A$, and computes the optimistic policy again. The aim is to ensure that $\hat{P}$ and $K$ are independent, so that we can use the concentration inequality on $(\hat{P} - P)V^*_{K,g}$.
> Our algorithm 3 (VALAE) works in this way: Algo 3 first uses Algo 1 to compute a set $K$ such that $K$ contains $S_L$ and $K$ is $(1+\epsilon)L$-controllable, then Algo 3 fixes the set $K$ and only focus on the policies restricted on $K$, i.e. it solves multi-goal SSP problem on $K$.
>
> >-Can you provide preliminary experiments to illustrate how the proposed algorithms behave on very simple MDP?
>
> We will consider adding some simulations in the final version. Thanks for your suggestion.
>
> >-Can you provide the time complexity and space complexity of VALAE algorithm?
>
> Time complexity: $\tilde{O}(TK^4A^3)$, where $T$ is the  sample complexity.
> The bottleneck is on the VISGO procedure. The proof is the same with Tarbouriech et al. 2020, Appendix G, where we substitute $S$ = $K$.
>
> Space complexity: $\tilde{O}(T+K^2A) = \tilde{O}(T)$, where $T$ is sample complexity. We note that we do not need $\tilde{O}(S^2A)$ space in prior work, where $S$ is the total number of states in the MDP.
> The bottleneck is on storing the empirical model $\widehat{P}$. First we fix $\mathcal{K}$ in a fixed round. We observe that for all $s \notin \mathcal{K}$, $V(s)$ are the same, and our algorithms only use the inner product $\widehat{P}V$ and variance $Var(\widehat{P}, P)$, thus we do not need to use $P_{s,a,s’}$ individually when $s’ \notin \mathcal{K}$, we need only to store $P_{s,a,s’}$ for $s’ \in \mathcal{K}$. When $\mathcal{K}$ changes, we can use the previous samples to recompute $\widehat{P}$.
>
> Specific Comments:
>
> >-Introduction: the sentence "This procedure also resembles some biological learning processes" is a bit cryptic.
>
> This argument comes from the paper “Autonomous Exploration For Navigating In MDPs”. We are happy to remove it.
>
> >-P1, note 1: It seems that there is an extra L in the big O.
>
> Yes, this is a typo.
>
> >-P4, end of the page: the learning framework is not completely clear: how does the agent interact with the MDP? when does the agent stop and output the policies (is T chosen by the agent)?
>
> The agent interact with MDP in this way: The agent knows its current state $s$ and the action space $A$. The agent does not know the model $P(s’|s,a)$ and cost function $c(s,a)$. Each time, the agent can choose an action $a$, and the agent will observe that it transit to a new state $s’$ with a cost $c$, where $s’$ and $c$ are revealed to the agent. The agent can stop and output the policies anytime when the agent thinks that it has collected enough samples, and thus $T$ is chosen by the agent.
>
> >-P5, end of the page: the sentence "Then for each action [..] \hat{c}(s,a)" is not clear at all.
>
> This is what we hope to do:
> We hope that for each state-action pair $(s,a)$ in $\mathcal{K} \times \mathcal{A}$, we can obtain $\phi$ samples of the form $(s,a,s’,c)$, hence our estimation $\hat{P}(s'|s,a)$ and $\hat{c}(s,a)$  are close enough to $P_{s,a,s’}$ and $c(s,a)$, respectively.
> We note that because $s$ is in $\mathcal{K}$, we have already known a policy $\pi_s$ that can arrive $s$ from $s_0$ within $L$ steps. Hence, to get a sample at state-action pair $(s,a)$, we need only to first execute $\pi_s$ to arrive state $s$, then execute action $a$.
>
> >-P6, Alg2, L5: typo "where x is defined".
>
> Yes, we can remove it.
>
> >-P8, top of the page: The first paragraph is hard to follow since we do not know at this point what are Lemma 15 and 16.
>
> In this paragraph, Lemma 15 means that the expected cost of the greedy policy $\tilde{\pi}$ is bounded by $2L$ ($\tilde{\pi}$ is the greedy policy over $Q$, where $Q$ is output of VISGO). Lemma 16 means that we can execute $\tilde{\pi}$ for $1/\epsilon^2$ times and compute the average cost, so that if the average cost of $\tilde{\pi}$ is smaller than $L + \epsilon$, with high probability the expected cost of $\tilde{\pi}$ is smaller than $L + 3\epsilon$.
> Thanks for mentioning this. We will further clarify in the final version.

---

### Decision · Program_Chairs · 2022-01-20

**Decision:**

Reject

**Comment:**

The paper studies an interesting problem, but as pointed out by reviewers, the presentation of the problem statement and contributions need to be improved.